# SFDIFF : DIFFUSION MODEL WITH SELF-GENERATION FOR PROBABILISTIC FORECASTING

## ABSTRACT

Diffusion models have emerged as an effective approach for time-series probabilistic forecasting, aiming to generate future observations based on historical data through a denoising process. In this paper, we introduce Self-Generation technique designed to enhance the performance of conditional generation in time-series forecasting. Self-Generation involves synthesizing not only future observations, but also historical data itself conditioned on the given historical context. While noise is often introduced during the observation process, our method can reduce the amount of noise in observed historical data, thereby enhancing forecasting accuracy. Additionally, to further boost forecasting performance, we incorporate classifier-free generation methods into conditional generation for time-series forecasting. In the experiment, we demonstrate that our method outperforms other condition generation methods.

## 1 INTRODUCTION

Time series forecasting is a critical problem in the fields of machine learning and deep learning, focusing on predicting future observations based on historical data. This process requires learning the relationship patterns between past and future data during training and, at inference time, reconstructing patterns that best fit the given historical data from the learned relationships. Time-series forecasting problems are essential across many domains, spanning fields such as physics, climate, healthcare, and finance (Lim & Zohren, 2021; Torres et al., 2021a; Masini et al., 2023).

Over the years, numerous deep learning methods have been proposed to address time-series forecasting problems (Lim & Zohren, 2021; Torres et al., 2021b; Miller et al., 2024). Among these, diffusion-based conditional generative methods have shown strong forecasting performance. (Rasul et al., 2021; Tashiro et al., 2021; Yan et al., 2021) These methods involve training neural networks to approximate the score values ($\nabla_{\mathbf{x}_t^{\text{pred}}} \log p(\mathbf{x}_t^{\text{pred}}|\mathbf{x}^{\text{hist}})$) on diffusion step $t$, where $p(\mathbf{x}^{\text{pred}}|\mathbf{x}^{\text{hist}})$ represents the conditional distribution of future observations given historical data. Using these trained networks, future observations are synthesized through a denoising process, including a reverse Stochastic Differential Equation (SDE) process or ancestral sampling, conditioned on the historical data.

However, due to measurement errors or the occurrence of anomalies, the conditional historical observations may contain noise, which can negatively impact forecasting performance (Rožanec et al., 2021). To address this, we propose Self-Generation, which extends the denoising process of score-based generation to both future observations and conditional historical data, effectively reducing the inherent noise in the conditional inputs. Specifically, our training objective is not to approximate $\nabla_{\mathbf{x}_t^{\text{pred}}} \log p(\mathbf{x}_t^{\text{pred}}|\mathbf{x}^{\text{hist}})$ but rather $\nabla_{\mathbf{x}_t^{\text{total}}} \log p(\mathbf{x}_t^{\text{total}}|\mathbf{x}^{\text{hist}})$, where $\mathbf{x}^{\text{total}}$ represents an union of two time-series $\mathbf{x}^{\text{pred}}$ and $\mathbf{x}^{\text{hist}}$.

Modeling the total sequence within the diffusion process has two major advantages. First, high-frequency anomalies in the data are effectively mitigated after a few forward diffusion steps (Choi et al., 2022; Yang et al., 2023). Second, during the generation process, predictions and conditions are interdependently generated. This interdependence enables the reverse diffusion process to minimize the impact of anomalies in the historical data, as predictions are generated using predominantly non-anomalous conditions and purified conditions informed by the generated predictions. To further leverage the noise-reduction capabilities of Self-Generation and focus on prediction generation,

we introduce an imbalanced weighting scheme in the loss function between the past and future components, demonstrating the importance of our careful training design.

To further enhance the forecasting performance of our score-based conditional generation, we integrate classifier-free generation introduced by Ho & Salimans (2022), with score-based conditional generation for time-series forecasting into a unified framework. Our results show that incorporating classifier-free methods into conditional generation significantly reduces errors on the most of datasets, with the only exception of Solar, and this positive impact is further amplified when combined with Self-Generation. Finally, in the experimental section, we demonstrate that our proposed method achieves state-of-the-art forecasting performance across 2 toy datasets and 5 real datasets, outperforming 12 baseline methods. To summarize, our contributions can be outlined as follows:

1. We propose Self-Generation as a novel approach for score-based conditional generation in time-series forecasting. Self-Generation reduces noise in conditional historical observations by synthesizing both future and historical observations through a denoising process, thereby enhancing forecasting performance.

2. We theoretically show how generating the entire time-series (rather than only the future part) within the diffusion process enables a noise-purification mechanism, and formally derive a corresponding total-sequence score-matching objective. In Section 3, we present Theorems 1–2 to justify this extended generation strategy and detail the resulting loss function, which emphasizes accurate predictions informed by denoised historical conditions.

3. To further enhance the forecasting performance of Self-Generation, we adapt the classifier-free generation approach to suit time-series forecasting scenarios.

4. Out of 2 toy examples and 5 real datasets, our score-based conditional generative method with Self-Generation achieves state-of-the-art performance in all cases, compared to 12 baselines, including methods based on variational autoencoder (VAE), diffusion, and gaussian process (GP).

## 2 PRELIMINARY AND PROBLEM STATEMENT

### 2.1 DIFFUSION MODELS

Generative models aim to synthesize realistic data, such as images, by learning the underlying probability distribution of the data (Oussidi & Elhassouny, 2018; Harshvardhan et al., 2020; Cao et al., 2024). Among various generative approaches, diffusion models have gained prominence outperforming generative adversarial network (GAN), in terms of generating high-quality images with more stable training (Dhariwal & Nichol, 2021; Song et al., 2020; Ho et al., 2020; Cao et al., 2024). Diffusion models operate through the following two-step process: i) **Noising** step, which means gradually adding noise to an image, transforming it into Gaussian noise, ii) **Denoising** step, which means recovering the original image from the noisy version, where the noise is sampled from a specific distribution, typically a normal distribution (Yang et al., 2023).

Initially, the denoising process was designed to reverse the noising process by adding noise in the opposite direction at each step. This process is derived from minimizing the Kullback-Leibler (KL) divergence between the joint probability of noising and denoising step, leading to an inequality involving the negative log-likelihood, similar to the variational autoencoder (VAE) framework. This approach is called Denoising Diffusion Probabilistic Models (DDPMs) (Ho et al., 2020).

Given original image $\mathbf{x} \sim p(\mathbf{x})$ and the length of noising and denoising step $T$, DDPMs add noise to the image according to the transition kernel: $p(\mathbf{x}_t|\mathbf{x}_{t-1}) = \mathcal{N}(\mathbf{x}_t; \sqrt{1 - \beta_t}\mathbf{x}_{t-1}, \beta_t\mathbf{I})$, where $t \in \{1, 2, ..., T\}$ and $\beta_t \in (0, 1)$ is a hyperparameter. With sufficiently large $T$, $\mathbf{x}_t$ converges to a normal distribution. DDPMs then train a corresponding learnable denoising kernel $p_\theta(\mathbf{x}_{t-1}|\mathbf{x}_t) = \mathcal{N}(\mathbf{x}_{t-1}; \mu_\theta(t, \mathbf{x}_t), \Sigma(t, \mathbf{x}_t))$, where the denoising process aims to reverse the added noise.

As a follow-up research, Song et al. (2020) have generalized diffusion models from discrete-time processes to continuous Stochastic Differential Equation (SDE) formulations, introducing Variance Exploding (VE), Variance Preserving (VP), and sub-VP processes. In this framework, the noising and denoising processes of diffusion models are reinterpreted as forward and reverse SDEs, respectively:

$$dx = \mathbf{f}(t, \mathbf{x})dt + g(t)d\mathbf{w}$$

$$dx = [\mathbf{f}(t, \mathbf{x}) - g(t)^2 \nabla_{\mathbf{x}} \log p_t(\mathbf{x})]dt + g(t)d\bar{\mathbf{w}}$$

, where $t \in [0, 1]$, $f$ is an affine and $\mathbf{w}, \bar{\mathbf{w}}$ represent forward and backward Brownian motion, respectively. Among these, the VP process is particularly notable for its connection to DDPMs, where: $\mathbf{f}(t, \mathbf{x}) = -\frac{1}{2}\beta(t)\mathbf{x}, g(t) = \sqrt{\beta(t)}$. They demonstrate that score based generative models train score network $s_\theta(\cdot, \cdot)$ to learn a gradient of log likelihood, score function, by using following score matching loss:

$$L_{SM}(\theta) = \mathbb{E}_{t,\mathbf{x}_t}[\lambda(t)||s_\theta(t, \mathbf{x}_t) - \nabla_{\mathbf{x}_t} \log p(\mathbf{x}_t)||^2],$$

where $\mathbf{x}_t \sim p(\mathbf{x}_t)$. However, directly using score matching loss is computationally prohibitive since calculating exact score function of $\mathbf{x}_t$ needs statistical method (Hyvärinen, 2005; Song et al., 2020). Thanks to specific formulation of $\mathbf{f}$ and $g$, we can derive a following denoising score matching loss, which can be calculated by using given formula (Vincent, 2011; Øksendal, 2014):

$$L_{DSM}(\theta) = \mathbb{E}_{t,\mathbf{x},\mathbf{x}_t}[\lambda(t)||s_\theta(t, \mathbf{x}_t) - \nabla_{\mathbf{x}_t} \log p(\mathbf{x}_t|\mathbf{x})||^2],$$

where $\mathbf{x} \sim p(\mathbf{x}), \mathbf{x}_t \sim p(\mathbf{x}_t|\mathbf{x})$. We can directly derive the equivalence between $L_{SM}(\theta)$ and $L_{DSM}(\theta)$ by considering the structure of the forward SDE. The drift term $\mathbf{f}(\cdot, \cdot)$ is affine and the diffusion term $g(\cdot)$ depends solely on the diffusion step. This results in the conditional probability $p(\mathbf{x}_t|\mathbf{x})$ being represented as a Gaussian distribution, $\mathcal{N}(\mathbf{x}_t; \mu_t(\mathbf{x}), \sigma_t)$ (Øksendal, 2014). Therefore, we can compute the gradient of log likelihood, $\nabla_{\mathbf{x}_t} \log p(\mathbf{x}_t|\mathbf{x})$, as: $\nabla_{\mathbf{x}_t} \log p(\mathbf{x}_t|\mathbf{x}) = -(\mathbf{x}_t - \mathbf{x})/\sigma_t^2 = -\epsilon/\sigma_t$, where the reparametrization trick is used on $\mathbf{x}_t = \mu_t(\mathbf{x}_t) + \sigma_t\epsilon$ and $\epsilon \sim \mathcal{N}(\mathbf{0}, \mathbf{I})$.

By linking the SDE and ODE formulations, Song et al. (2021) proposed setting $\lambda(t) = g^2(t)$ to ensure the following inequality:

$$-\mathbb{E}_{\mathbf{x}}[\log p(\mathbf{x})] \leq L_{SM}(\theta) + C_1,$$

where $L_{SM}(\theta) = \mathbb{E}_{t,\mathbf{x}_t}\left[g^2(t)||s_\theta(t, \mathbf{x}_t) - \nabla_{\mathbf{x}_t} \log p(\mathbf{x}_t)||_2^2\right]$ and $C_1$ is a constant. Based on this, we adopt $g^2(\cdot)$ as the default weighting in our experiments.

Once the score network is trained, diffusion models proceed with the denoising step. At this stage, there are two main sampling strategies: the predictor-corrector (PC) sampler and a deterministic sampler based on the probability flow ordinary differential equation (ODE). In here, we explain PC sampler that is used in our experiment. The PC sampler works by first estimating the next step using a known numerical SDE solver, which is called *predictor*. Then refining the estimate with a score-based MCMC strategy, which is named of *corrector*. A representative example of predictor is an Euler-Maruyama sampling predictor, which is a discretization of backward SDE:

$$\mathbf{x}_{t-1} = [\mathbf{f}(t, \mathbf{x}_t) - g(t)^2 s_\theta(t, \mathbf{x}_t)]\Delta t + g(t)\Delta w,$$

, where $t \in [1, 0]$, $\Delta t$ is a time interval and $\Delta w \sim \mathcal{N}(\mathbf{0}, \Delta t\mathbf{I})$.

Song et al. (2020) achieved state-of-the-art results through extensive hyperparameter tuning of various SDEs, predictors and correctors. However, for our experiments, we adopt the VP SDE and use an Euler-Maruyama sampling predictor without corrector, which is a default setting of it (Song et al., 2020). This allows us to isolate the performance of SFdiff from other factors, ensuring that other control variables remain fixed.

## 2.2 TIME-SERIES FORECASTING

Time-series forecasting involves predicting future values based on historical data (Lim & Zohren, 2021; Torres et al., 2021b; Miller et al., 2024). Specifically, given a historical sequence $\mathbf{x}^{1:N}$, the task is to forecast the future sequence $\mathbf{x}^{N+1:N+T}$, where $N$ represents the length of the historical data, and $T$ represents the length of the prediction. Each data point $\mathbf{x}$ belongs to $\mathbb{R}^d$. For clarity, we define $\mathbf{x}^{\text{hist}}$ by a sequence of history, $\mathbf{x}^{1:N}$, $\mathbf{x}^{\text{pred}}$ by a future values, $\mathbf{x}^{N+1:N+T}$, and $\mathbf{x}^{\text{total}}$, a total sequence $\mathbf{x}^{1:N+T}$. Time-series forecasting has been widely researched improve the accuracy of future predictions. However, the complex, intertwined characteristics of time-series data make it difficult to fully capture and understand its underlying patterns.

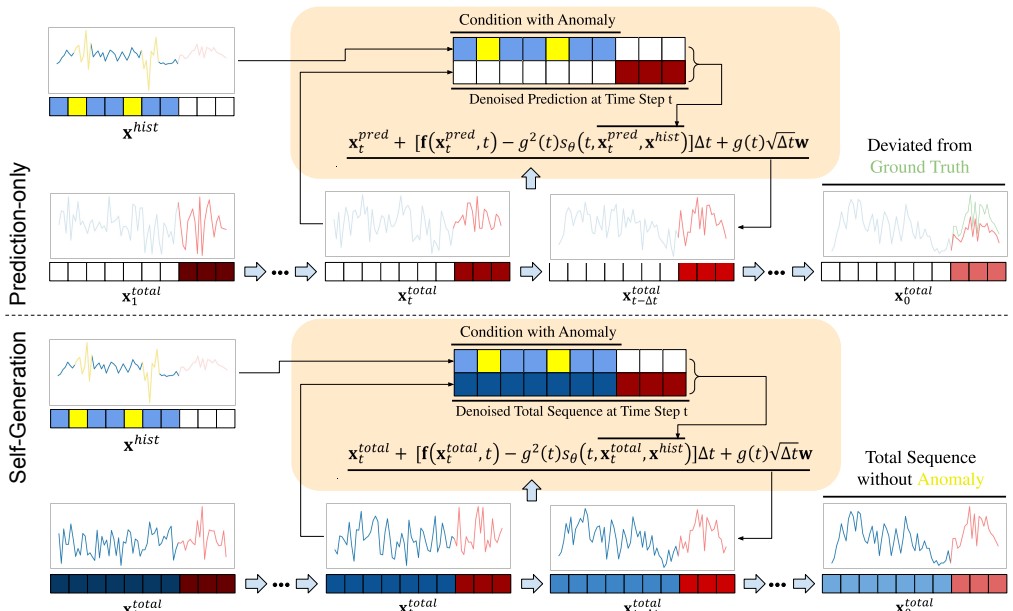

Figure 1: **Overall visualization of the sampling process of SFdiff.** Comparison between prediction-only generation (top) and Self-Generation (bottom). Even when anomalies (highlighted in yellow) exist in the historical data, SFdiff iteratively applies reverse diffusion to produce an anomaly-free sequence containing both historical and robust future observations, whereas prediction-only generation deviates from the ground truth. Although reverse SDE is shown here, other sampling strategies can be used interchangeably.

To address this challenge, researchers have increasingly turned to generative models, which aim to model the conditional likelihood of time-series data and provide a more comprehensive understanding of its structure. As a result, many time-series diffusion models were appeared, which generally aim to learn conditional distribution of prediction given history sequence, $p(\mathbf{x}^{\mathrm{pred}}|\mathbf{x}^{\mathrm{hist}})$ (Rasul et al., 2021; Tashiro et al., 2021). We provide a detailed explanation of their contributions and the rationale behind their target selection in Section 6. Therefore, those who apply DDPM methods to forecasting problem optimize the following equation:

$$L_{SM}^{\mathrm{pred}}(\theta) = \mathbb{E}_{t,\mathbf{x}_t^{\mathrm{pred}}}\big[\lambda(t)\|s_\theta(t,\mathbf{x}_t^{\mathrm{pred}},\mathbf{x}^{\mathrm{hist}}) - \nabla_{\mathbf{x}_t^{\mathrm{pred}}}\log p(\mathbf{x}_t^{\mathrm{pred}}|\mathbf{x}^{\mathrm{hist}})\|_2^2\big].$$

## 3 PROPOSED METHOD

In this section, we analyze existing prediction methods and propose a novel Self-Generation approach that reconstructs the entire sequence consisting of purified condition given noised history data.

### 3.1 DIFFUSION MODEL WITH SELF-GENERATION

Current methods train diffusion models by optimizing the conditional probability $p(\mathbf{x}^{\mathrm{pred}}|\mathbf{x}^{\mathrm{hist}})$ through well-known DDPM loss (Rasul et al., 2021; Tashiro et al., 2021; Kollovieh et al., 2023b):

$$L_{DDPM}^{pred}(\theta) = \mathbb{E}_{t,\epsilon,\mathbf{x}^{pred}}[\lambda(t)\|\epsilon - \epsilon_\theta(t,\mathbf{x}_t^{pred},\mathbf{x}^{hist})\|^2]$$

Song et al. (2021) proved that the target value of DDPM ($\log p(\mathbf{x}^{\mathrm{pred}}|\mathbf{x}^{\mathrm{hist}})$) can be optimized in the perspective of score-based approach:

$$-\mathbb{E}_{\mathbf{x}^{\mathrm{total}}}\log p(\mathbf{x}^{\mathrm{pred}}|\mathbf{x}^{\mathrm{hist}}) \le \frac{1}{2}\cdot L_{SM}^{\mathrm{pred}}(\theta) + C_1,$$

where

$$L_{SM}^{\mathrm{pred}}(\theta) = \mathbb{E}_{t,\mathbf{x}^{\mathrm{hist}},\mathbf{x}_t^{\mathrm{pred}}}\big[\lambda(t)\|s_\theta(t,\mathbf{x}_t^{\mathrm{pred}},\mathbf{x}^{\mathrm{hist}}) - \nabla_{\mathbf{x}_t^{\mathrm{pred}}}\log p(\mathbf{x}_t^{\mathrm{pred}}|\mathbf{x}^{\mathrm{hist}})\|_2^2\big].$$

While intuitive, this approach is sensitive to noisy conditions: if $\mathbf{x}^{\text{hist}}$ contains adversarial noise, the resulting diffusion model may produce degraded samples. To address this, we propose SFdiff : Diffusion Model with Self-Generation for Probabilistic Forecasting, which generates the entire time-series instead of only synthesizing prediction sequence. SFdiff learns mathematically same conditional distribution $p(\mathbf{x}^{\text{total}}|\mathbf{x}^{\text{hist}}) = p(\mathbf{x}^{\text{pred}}|\mathbf{x}^{\text{hist}})$ by optimizing the following inequality:

$$-\mathbb{E}_{\mathbf{x}^{\text{total}}} \log p(\mathbf{x}^{\text{total}}|\mathbf{x}^{\text{hist}}) \leq \frac{1}{2} \cdot L_{SM}^{\text{total}}(\theta) + C_1,$$

where

$$L_{SM}^{\text{total}}(\theta) = \mathbb{E}_{t,\mathbf{x}^{\text{hist}},\mathbf{x}_t^{\text{total}}} \left[ \lambda(t) \| s_\theta(t, \mathbf{x}_t^{\text{total}}, \mathbf{x}^{\text{hist}}) - \nabla_{\mathbf{x}_t^{\text{total}}} \log p(\mathbf{x}_t^{\text{total}}|\mathbf{x}^{\text{hist}}) \|_2^2 \right].$$

Generating the total time-series offers two key advantages:

1. **Noise Purification:** Diffusion models inherently denoise conditions. By learning the conditional distribution of the total time-series, the model can generate a purified sequence consistent with the full distribution.

2. **Interdependent Predictions:** During total sequence generation, predictions are both influenced by and influence historical data through the diffusion process.

The following theorem demonstrates how the Self-Generation preserve robustness between noisy and clean conditions:

**Theorem 3.1.** *Assume (A1)–(A3) in Appendix A. Let $H(t) := \int_t^1 g(s)^2\, ds$ and $G := H(0) = \int_0^1 g(s)^2\, ds > 0$. Then for noised input condition $\mathbf{c}_s \equiv \mathbf{x}_s^{hist}$, the upper bound of difference between prediction and ground truth $\|\mathbf{x}_0^{pred} - \mathbf{x}_0^{pred'}\|$ are*

$$\textit{(prediction-only)} \quad L \int_0^1 g(s)^2 \, \|\mathbf{c}_s - \mathbf{c}'_s\|\, ds \;\leq\; L\, G \sup_{s \in [0,1]} \|\mathbf{c}_s - \mathbf{c}'_s\|, \tag{1}$$

$$\textit{(total-sequence)} \quad L \int_0^1 g(s)^2\, e^{-m_x H(s)} \|\mathbf{c}_s - \mathbf{c}'_s\|\, ds \;=\; \frac{L}{m_x}\big(1 - e^{-m_x G}\big) \sup_s \|\mathbf{c}_s - \mathbf{c}'_s\|. \tag{2}$$

*Consequently,*

$$\frac{L}{m_x}\big(1 - e^{-m_x G}\big) \;<\; L\, G \quad \Rightarrow \quad \sup \|\mathbf{x}_0^{pred} - \mathbf{x}_0^{pred'}\|_{\textit{total-sequence}} \;<\; \sup \|\mathbf{x}_0^{pred} - \mathbf{x}_0^{pred'}\|_{\textit{pred-only}}.$$

*Thus the* total-sequence *conditional score produces forecasts with smaller sensitivity bound to condition perturbations than the* prediction-only *score.*

On the Theorem 3.1, we distinguish $\mathbf{x}^{\text{hist}}$ from condition and total sequence by denoting that of condition as $\mathbf{c}$. The above theorem shows that when noise is injected to history data, under (A1)–(A3), Self-Generation has a strictly smaller robustness constant than prediction-only; thus any Lipschitz evaluation functional inherits a no-worse (and, under nonzero perturbations, strictly better) worst-case sensitivity.

On the other hand, It is well known that directly computing $L_{SM}^{\text{pred}}(\theta)$ and $L_{SM}^{\text{total}}(\theta)$ is computationally prohibitive due to the need for statistical methods (Hyvärinen, 2005; Song et al., 2020). Therefore, we derive the denoising score-matching losses to train the score network $s_\theta$ and guarantee its convergence:

**Theorem 3.2.** *For each $L_{SM}^{pred}(\theta)$ and $L_{SM}^{total}(\theta)$, its denoising score matching are represented as follows:*

$$L_{DSM}^{pred}(\theta) = \mathbb{E}_{t,\mathbf{x}^{total},\mathbf{x}_t^{total}}[\lambda(t) || s_\theta(t, \mathbf{x}_t^{pred}, \mathbf{x}^{hist}) - \nabla_{\mathbf{x}_t^{pred}} log p(\mathbf{x}_t^{total}|\mathbf{x}^{total}) ||_2^2]$$

$$L_{DSM}^{total}(\theta) = \mathbb{E}_{t,\mathbf{x}^{total},\mathbf{x}_t^{total}}[\lambda(t) || s_\theta(t, \mathbf{x}_t^{total}, \mathbf{x}^{hist}) - \nabla_{\mathbf{x}_t^{total}} log p(\mathbf{x}_t^{total}|\mathbf{x}^{total}) ||_2^2]$$

*Therefore, these models aim same conditional score function since $\nabla_{\mathbf{x}_t^{total}} log p(\mathbf{x}_t^{total}|\mathbf{x}^{total}) = \nabla_{[\mathbf{x}_t^{hist},\mathbf{x}_t^{pred}]} log p(\mathbf{x}_t^{total}|\mathbf{x}^{total})$.*

Beyond using $L_{DSM}^{\text{total}}(\theta)$, we place additional emphasis on the prediction portion of the sequence. In designing SFdiff, we aim to ensure that it generates a predictive sequence that takes past history into account but is not overly dominated by historical values. To achieve this balance, we introduce a hyperparameter $\gamma$ to control the influence of the past history. The exact loss function is then defined as:

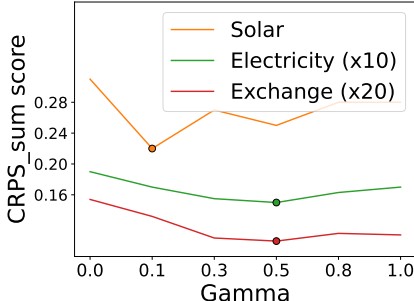

$$l(\theta) = ||s_\theta(t, \mathbf{x}_t^{\text{total}}, \mathbf{x}^{\text{hist}}) - \nabla_{\mathbf{x}_t^{\text{total}}} \log p(\mathbf{x}_t^{\text{total}}|\mathbf{x}^{\text{total}})||^2,$$
$$L(\theta) = \mathbb{E}_{t, \mathbf{x}^{\text{total}}, \mathbf{x}_t^{\text{total}}}[\lambda(t)||\gamma \mathbf{m} \otimes l(\theta) + (1 - \mathbf{m}) \otimes l(\theta)||_1]$$

Figure 2: $CRPS_{sum}$ on different $\gamma$.

, where $\otimes$ is a hadamard product and $\mathbf{m} = \{x_{ij}\}_{(N+T) \times d}$ is a mask vector that $x_{ij} = 1$ if $i \leq N$ and $0$ otherwise, dividing the past and future elements in our loss function.

### 3.2 TRAINING AND INFERENCE

For training SFdiff, we employ the Variance Preserving (VP) SDE (Song et al., 2020), which generalizes existing DDPM-based methods (Rasul et al., 2021; Tashiro et al., 2021) to compute $\nabla_{\mathbf{x}_t} \log p(\mathbf{x}_t|\mathbf{x})$. In this phase, a key aspect is controlling the parameter $\gamma$. Setting $\gamma = 0.0$ (equivalent to DDPM) ignores past information, while $\gamma = 1.0$ fully incorporates it. Neither extreme is optimal. As shown in Figure 2 and Figure 3, SFdiff achieves the best results on the Solar and Electricity datasets when $\gamma = 0.1$ and $\gamma = 0.5$, respectively.

After training the model, we generate total sequence from history condition by using well-known PC sampling procedure and its default setting (Song et al., 2020). Furthermore, to facilitate Self-Generation technique, we adapt classifier-free guidance(CFG) to our framework. However, up to our survey, CFG is confined to DDPM and there was no adaptation of CFG to score-based diffusion models. Therefore, we briefly introduced CFG in Section C and now explain its usage in SFdiff.

From a score matching perspective, CFG using score function can be understood as $\nabla_{\mathbf{x}_t} \log \tilde{p}(\mathbf{x}_t|\mathbf{c}) = \nabla_{\mathbf{x}_t} \log p(\mathbf{x}_t|\mathbf{c}) + w\nabla_{\mathbf{x}_t} \log p(\mathbf{c}|\mathbf{x}_t) = \nabla_{\mathbf{x}_t} \log p(\mathbf{x}_t|\mathbf{c}) + w\nabla_{\mathbf{x}_t}(\log p(\mathbf{x}_t|\mathbf{c}) - \log p(\mathbf{x}_t)) = (1 + w)\nabla_{\mathbf{x}_t} \log p(\mathbf{x}_t|\mathbf{c}) - w\nabla_{\mathbf{x}_t} \log p(\mathbf{x}_t)$. And this formulation leads to the generalized score function used in CFG: $\tilde{s}_\theta(\mathbf{x}_t, \mathbf{c}) = (1 + w)s_\theta(\mathbf{x}_t, \mathbf{c}) - ws_\theta(\mathbf{x}_t, \mathbf{0})$, where $\mathbf{0}$ means zero padding. We use this generalized CFG sampling. As the formulation shows, CFG should train both conditional and unconditional sampling to single model. In line with Ho & Salimans (2022), we adopt a proportional training strategy, where with probability $p_{\text{cond}}$ (setting 0.2 as default value), the model trains the conditional score network $s_\theta(\mathbf{x}_t, \mathbf{c})$, and with probability $1 - p_{\text{cond}}$, it trains the unconditional score network $s_\theta(\mathbf{x}_t, \mathbf{0})$.

Comparing Self-Generation with prior prediction-only generation methods reveals interesting insights. Intuitively, CFG applied to prediction generation with potentially noisy conditions may amplify undesirable influences, degrading performance. In contrast, Self-Generation benefits from CFG by generating predictions jointly with a denoised historical sequence. As shown in Table 1, CFG negatively impacts prediction-only generation by exacerbating noise-related effects.

Table 1: $CRPS_{sum}$ comparison between prediction CFG results on prediction generation and total generation.

|  | Exchange | Electricity | Solar |
|---|---|---|---|
| $L^{\text{pred}}$ | .006±.001 | .021±.001 | .287±.020 |
| $L_{CFG}^{\text{pred}}$ | .008±.001 | .026±.001 | .451±.011 |
| $L^{\text{total}}$ | .006±.000 | .018±.001 | .250±.007 |
| $L_{CFG}^{\text{total}}$ | .005±.000 | .015±.000 | .277±.006 |

However, combined with Table 2, Self-Generation yields overall improved results, with minimal performance degradation observed for the Solar dataset, which is the only reduced result among dataset.

## 4 EXPERIMENTS

In this section, we present the results of experiments conducted to evaluate the performance of our proposed model.

### 4.0.1 EXPERIMENTAL SETUPS

Our experiments consist of two stages: (1) assessing whether our generative framework effectively reduces noise in the conditions using toy datasets, and (2) evaluating our model's performance on real-world time-series datasets.

To verify noise reduction in the conditions, we utilize two toy datasets: the 2D oscillator ODE and the 3D harmonic ODE. The corresponding ODE formulations are as follows:

$$
\begin{bmatrix} \frac{dx}{dt} \\ \frac{dy}{dt} \end{bmatrix} = \begin{bmatrix} y \\ y(1-x)^2 - x \end{bmatrix} \text{ and } \begin{bmatrix} \frac{dx}{dt} \\ \frac{dy}{dt} \\ \frac{dz}{dt} \end{bmatrix} = \begin{bmatrix} -y \\ x-z \\ y \end{bmatrix},
$$

where the initial conditions are $[2.0, 0.0]$ and $[0.5, 0.5, 0.5]$, respectively. We generate the ODE trajectories using the well-known `scipy` package with a time interval of 0.1.

For real dataset experiments, we use our model on 5 widely-used time-series forecasting datasets: Exchange (Lai et al., 2017), Solar (Lai et al., 2017), Electricity[1], Taxi[2], Wikipedia[3]. We give detailed description of these datasets in Table 5, including dimension, total number of timesteps, domain and frequency data of each dataset. We also report hyperparameters setting in Table 5: the history and prediction lengths, the number of diffusion steps, and the number of iterations, etc. Here, we point out that we follow the common practice of training based on iteration count and saving checkpoints every 5,000 steps, as done in other diffusion models (Ho et al., 2020; Song et al., 2020).

After training our model on the selected real datasets, we evaluate its performance against a wide range of baseline models. These baselines include: i) classical multivariate methods such as VAR, VAR-Lasso (Lütkepohl, 2005), GARCH (van der Weide, 2002), and VES (Hyndman et al., 2008); ii) RNN-based methods like Vec-LSTM-ind-scaling, Vec-LSTM-lowrank-Copula, GP-scaling, and GP-Copula (Salinas et al., 2019); iii) Transformer-based models, specifically Transformer-MAF (Rasul et al., 2020); and iv) VAE and diffusion-based models, including KVAE (Fraccaro et al., 2017), MG-TSD (Fan et al., 2024), TimeGrad (Rasul et al., 2021), and CSDI (Tashiro et al., 2021). A description of these baseline models can be found in Appendix D.

For evaluation, we use the sum of continuous ranked probability score ($CRPS_{sum}$), a widely recognized metric for probabilistic forecasting. CRPS measures the compatibility between the cumulative distribution function (CDF) $F$ and an observation $x$ as $CRPS(F, x) = \int (F(z) - \mathbb{I}(x \leq z))^2 dz$, where $\mathbb{I}$ is an indicator function. To approximate CDF, we use an empirically estimated CDF $\hat{F} = \frac{1}{N} \sum_{i=1}^{N} \mathbb{I}(x_i \leq z)$, where $x_i$ are samples from $F$. Then we compute the sum of CRPS over all features, denoted as $CRPS_{sum}$,

$$
CRPS_{sum}(F, x) = \frac{CRPS(F, \sum_i x_{i,t})}{\sum_{i,t} |x_{i,t}|}
$$

, where $\sum_{i,t} |x_{i,t}|$ means the summation of all target features at time $t$. For other detailed descriptions of experimental setup, we refer to Section B.

### 4.1 EXPERIMENTS ON TOY DATASETS

We utilize two toy datasets: the 2D oscillator dataset and the 3D harmonic dataset. During training, we augment the given trajectories by randomly adding noise sampled from $\mathcal{N}(0, \frac{1}{2}\mathbf{I})$ to half of the trajectory to promote robust training. Each trajectory is uniformly divided into segments of length 72. For testing, we introduce more intense noise, sampled from $\mathcal{N}(0, \mathbf{I})$, to $\frac{1}{8}$ of the condition portion of the test samples to evaluate the purification effectiveness of our model.

Figure 3 illustrates that our model effectively purifies noisy conditions, significantly reducing large anomalous values. Notably, the bottom row of the figure shows that the parameter $\gamma$ in our loss

---

[1]https://archive.ics.uci.edu/ml/datasets/ElectricityLoadDiagrams20112014

[2]https://www1.nyc.gov/site/tlc/about/tlc-trip-record-data.page

[3]https://github.com/mbohlkeschneider/gluon-ts/tree/mv_release/datasets

Table 2: $CRPS_{sum}$ results on evaluation datasets. The best scores are in boldface.

|  | Exchange | Solar | Electricity | Taxi | Wiki |
|---|---|---|---|---|---|
| VES | **.005**±**.000** | .900±.003 | .880±.004 | - | - |
| VAR | **.005**±**.000** | .830±.006 | .039±.001 | - | - |
| VAR-Lasso | .012±.000 | .510±.006 | .025±.000 | - | 3.10±.004 |
| GARCH | .023±.000 | .880±.002 | .190±.001 | - | - |
| KVAE | .014±.002 | .340±.025 | .051±.019 | - | .095±.012 |
| Vec-LSTM ind-scaling | .008±.001 | .391±.017 | .025±.001 | .506±.005 | .133±.002 |
| Vec-LSTM low-copula | .007±.000 | .319±.011 | .064±.008 | .326±.007 | .241±.033 |
| GP scaling | .009±.000 | .368±.012 | .022±.000 | .183±.395 | 1.48±1.03 |
| GP copula | .007±.000 | .337±.024 | .025±.002 | .208±.183 | .086±.004 |
| Transformer MAF | .005±.003 | .301±.014 | .021±.000 | .179±.002 | .063±.003 |
| MG-TSD | .007±.002 | .308±.010 | **.015**±**.002** | .116±.013 | .053±.005 |
| TimeGrad | .006±.001 | .287±.020 | .021±.001 | .114±.020 | .049±.002 |
| CSDI | .007±.001 | .298±.004 | .017±.000 | .123±.003 | .047±.003 |
| SFdiff | .006±.000 | **.250**±**.007** | .018±.001 | .122±.001 | .052±.000 |
| SFdiff-CFG | **.005**±**.000** | .277±.006 | **.015**±**.000** | **.092**±**.001** | **.046**±**.001** |

function plays a crucial role in achieving successful purification. When $\gamma \approx 1$, the synthesized time-series closely follow the noisy conditions, whereas when $\gamma \approx 0$, the model struggles to generate meaningful conditions. Thus, controlling $\gamma$ is essential not only for generating accurate predictions but also for mitigating out-of-distribution values, a process we refer to as purification.

## 4.2 EXPERIMENTS ON REAL DATASETS

We present the $CRPS_{sum}$ performance of SFdiff and other baseline models in Table 2. We evaluate SFdiff with 5 different seeds, and we report both the mean and standard deviation. As shown in the table, SFdiff consistently outperforms all competing models across every dataset, including other diffusion-based forecasting models. Notably, while diffusion-based forecasting models like TimeGrad and CSDI perform comparably on certain datasets, SFdiff consistently delivers superior results across a wide range of data complexities, from relatively low-dimensional datasets (e.g., Exchange) to high-dimensional ones (e.g., Wiki).

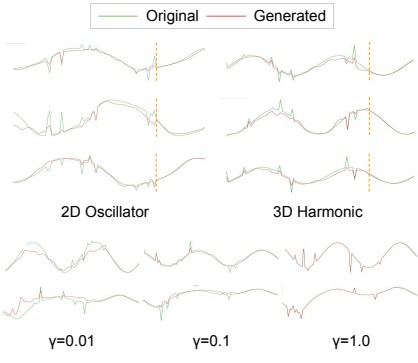

Figure 3: Generated total time-series. The orange dots divide history and prediction part.

## 5 ABLATION EXPERIMENTS

In this section, we conduct sensitivity studies about weight of classifier-free guidance. As shown in the Table 3, CFG results getting deteriorated as the weight of CFG ($w$) getting stronger.

Next, we present ablation studies conducted across several datasets to analyze the impact of varying the diffusion steps in SFdiff. We experiment with different numbers of diffusion steps: 50, 100, 200, 250, 500, and report the corresponding $CRPS_{sum}$ results.

|  | Original | CFG$_{0.01}$ | CFG$_{0.1}$ |
|---|---|---|---|
| Exchange | .006±.000 | **.005**±**.000** | .006±.000 |
| Electricity | .018±.001 | **.015**±**.000** | .016±.000 |
| Solar | **.250**±**.007** | .277±.006 | .300±.002 |

Table 3: Detailed dataset descriptions.

As indicated by the results, there are optimal "sweet spots" for the number of steps depending on the dataset. For example, SFdiff requires relatively fewer diffusion steps on datasets like Exchange and Electricity, whereas it benefits from higher steps on the Solar dataset to achieve the best performance. However, A large number of diffusion steps increases

Table 4: Results of ablation study varying the number of sampling steps

|  | 50 | 100 | 200 | 250 | 500 |
|---|---|---|---|---|---|
| Exchange | 0057±.0003 | **.0054±.0002** | .0057±.0002 | .0059±.0004 | .0057±.0002 |
| Electricity | .0168±.0003 | **.0165±.0005** | .0168±.0007 | .0166±.0005 | .0166±.0002 |
| Solar | .4540±.0125 | .2829±.0090 | .2501±.0070 | .2313±.0059 | **.2155±.0089** |

sampling time; we select a practical step budget (Table 5 in Appendix B) to balance accuracy and latency.

We also point out that a notable distinction of SFdiff, compared to other diffusion-based forecasting models such as CSDI (Tashiro et al., 2021) and TimeGrad (Rasul et al., 2021), is its ability to adjust the number of sampling steps without the need for additional training at each specific step. This flexibility offers a significant advantage, as it allows SFdiff to adapt more efficiently across varying datasets and conditions, no retraining cost, but sampling scales with steps.

## 6 RELATED WORK

This section briefly reviews diffusion-based time-series forecasting models, categorizing them based on their target score objectives.

Existing diffusion-based forecasting models are broadly divided into two categories: models targeting the prediction sequence score function, $\nabla_{\mathbf{x}_t^{\text{pred}}} \log p(\mathbf{x}_t^{\text{pred}}|\mathbf{x}^{\text{hist}})$, and those modeling the entire sequence score, $\nabla_{\mathbf{x}_t^{\text{total}}} \log p(\mathbf{x}_t^{\text{total}}|\mathbf{x}^{\text{hist}})$.

TimeGrad (Rasul et al., 2021) and CSDI (Tashiro et al., 2021) belong to the first category. TimeGrad generates predictions autoregressively, predicting one step ahead iteratively, whereas CSDI generates the entire prediction sequence in a single step. Although one-shot generation can be efficient, it may introduce higher variance in samples, prompting CSDI to stabilize performance by averaging multiple samples.

In the second category, models like TSDiff (Kollovieh et al., 2023b) [4] generate the complete sequence, leveraging history-guided sampling to enhance conditional generation. Additionally, Lim et al. (2023) and Lim et al. (2024) propose autoregressive generation in a latent space to handle irregularly sampled data effectively, offering improved modeling of complex time dependencies.

Our proposed method, SFdiff, combines advantages from both categories, integrating predictive accuracy and guidance mechanisms within a unified framework for robust and flexible time-series forecasting.

## 7 CONCLUSION

We propose the Self-Generation framework, leveraging diffusion models to robustly forecast time-series data despite anomalous inputs. Self-Generation effectively purifies noisy conditions by generating the entire sequence, balancing historical and future components. Moreover, we introduce classifier-free guidance into diffusion-based forecasting, significantly enhancing predictive accuracy. Extensive experiments demonstrate our model consistently outperforms 12 baselines across two toy examples and five real-world datasets. **Limitations**. While our approach achieves state-of-the-art forecasting performance, it requires careful tuning of the hyperparameter $\gamma$ to balance historical and predictive sequences effectively. Furthermore, the computational cost associated with diffusion models can be substantial, especially for large-scale applications. **Societal Impacts**. Our work presents positive societal impacts by improving predictive accuracy in critical domains such as healthcare and finance, potentially aiding better-informed decisions. However, misuse in sensitive areas, such as privacy-sensitive data forecasting, might lead to ethical concerns. **Safeguards**. Since our work primarily focuses on synthetic and publicly available datasets, specific safeguards for high-risk misuse scenarios were not required. However, we emphasize careful ethical consideration for future extensions involving sensitive or confidential data.

---

[4]We omit TSDiff on our main paper, since SFdiff aims to multivariate generation while TSDiff targets univariate generation. We further investigate TSDiff on Appendix E.

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

## A  DETAILED PROOF

In this section, we give detailed proof of Theorems.

### A.1  ROBUSTNESS OF SELF-GENERATION

We assume the generative structure

$$p(\mathbf{x}^{\text{hist}}, \mathbf{x}^{\text{pred}}, \tilde{\mathbf{x}}^{\text{hist}}) = p(\mathbf{x}^{\text{hist}}) \, p(\mathbf{x}^{\text{pred}} \mid \mathbf{x}^{\text{hist}}) \, p(\tilde{\mathbf{x}}^{\text{hist}} \mid \mathbf{x}^{\text{hist}})$$

and apply independent VP forward kernels to $(\mathbf{x}^{\text{hist}}, \mathbf{x}^{\text{pred}}, \tilde{\mathbf{x}}^{\text{hist}})$ to obtain $(\mathbf{x}_t^{\text{hist}}, \mathbf{x}_t^{\text{pred}}, \mathbf{c}_t)$ with $\mathbf{c}_t \equiv \tilde{\mathbf{x}}_t^{\text{hist}}$. Throughout we abbreviate $p_t(\cdot) = p(\cdot$ at time $t)$.

**Assumption A.1.**  (A1) (*Lipschitz-in-history score for the future*) Define $\phi_{\mathbf{h}}(\mathbf{p}) := \nabla_{\mathbf{p}} \log p_t(\mathbf{p} \mid \mathbf{h})$. There exists $L > 0$ such that $\|\phi_{\mathbf{h}}(\mathbf{p}) - \phi_{\mathbf{h}'}(\mathbf{p})\| \le L \|\mathbf{h} - \mathbf{h}'\|$ uniformly in $(\mathbf{h}, \mathbf{h}', \mathbf{p}, t)$.

(A2) (*Strong log-concavity in $\mathbf{x}^{hist}$*) $-\log p_t(\mathbf{x}^{\text{hist}} \mid \mathbf{c})$ is $m_x$-strongly convex in $\mathbf{x}^{\text{hist}}$, uniformly over $(\mathbf{c}, t)$; i.e.,

$$\left(\nabla_{\mathbf{x}^{\text{hist}}} \log p_t(\mathbf{x}^{\text{hist}} \mid \mathbf{c}) - \nabla_{\mathbf{x}^{\text{hist}}} \log p_t(\mathbf{x}^{\text{hist}'} \mid \mathbf{c})\right)^\top (\mathbf{x}^{\text{hist}} - \mathbf{x}^{\text{hist}'}) \le -m_x \|\mathbf{x}^{\text{hist}} - \mathbf{x}^{\text{hist}'}\|^2.$$

(A3) (*Coercivity in $\mathbf{x}^{pred}$*) $-\log p_t(\mathbf{x}^{\text{pred}} \mid \mathbf{x}^{\text{hist}})$ is $m_y$-strongly convex in $\mathbf{x}^{\text{pred}}$ uniformly in $(\mathbf{x}^{\text{hist}}, t)$.

Assumption (A1) covers conditional exponentials (e.g., conditionally Gaussian models), and (A2)–(A3) are standard to ensure contractive reverse flows.[5] These assumptions are satisfied by typical conditionally log-concave families (e.g., linear/affine–Gaussian and more general exponential families with Lipschitz natural parameters) and by neural score models whose Jacobians are Lipschitz/sector-bounded via spectral or weight normalization.

**Lemma A.2.**  *For any $t \in (0, 1]$,*

$$p_t(\mathbf{x}_t^{hist}, \mathbf{x}_t^{pred} \mid \mathbf{c}_t) = p_t(\mathbf{x}_t^{hist} \mid \mathbf{c}_t) \cdot p_t(\mathbf{x}_t^{pred} \mid \mathbf{x}_t^{hist}),$$

*hence $\mathbf{x}_t^{pred} \perp \mathbf{c}_t \mid \mathbf{x}_t^{hist}$.*

*Proof.* By construction $p(\mathbf{x}^{\text{hist}}, \mathbf{x}^{\text{pred}}, \tilde{\mathbf{x}}^{\text{hist}}) = p(\mathbf{x}^{\text{hist}}) \, p(\mathbf{x}^{\text{pred}} \mid \mathbf{x}^{\text{hist}}) \, p(\tilde{\mathbf{x}}^{\text{hist}} \mid \mathbf{x}^{\text{hist}})$, and the VP forward kernels act independently on $(X, Y, \widetilde{X})$. The Markov property yields $C_t \perp Y_t \mid X_t$ and thus the factorization $p_t(\mathbf{x}_t^{\text{hist}}, \mathbf{x}_t^{\text{pred}} \mid \mathbf{c}_t) = p_t(\mathbf{x}_t^{\text{hist}} \mid \mathbf{c}_t) \cdot p_t(\mathbf{x}_t^{\text{pred}} \mid \mathbf{x}_t^{\text{hist}})$.  □

From Lemma A.2,

$$\nabla_{\mathbf{x}_t^{\text{pred}}} \log p_t(\mathbf{x}_t^{\text{hist}}, \mathbf{x}_t^{\text{pred}} \mid \mathbf{c}_t) = \nabla_{\mathbf{x}_t^{\text{pred}}} \log p_t(\mathbf{x}_t^{\text{pred}} \mid \mathbf{x}_t^{\text{hist}}),$$

$$\nabla_{\mathbf{x}_t^{\text{hist}}} \log p_t(\mathbf{x}_t^{\text{hist}}, \mathbf{x}_t^{\text{pred}} \mid \mathbf{c}_t) = \nabla_{\mathbf{x}_t^{\text{hist}}} \log p_t(\mathbf{x}_t^{\text{hist}} \mid \mathbf{c}_t) + \nabla_{\mathbf{x}_t^{\text{hist}}} \log p_t(\mathbf{x}_t^{\text{pred}} \mid \mathbf{x}_t^{\text{hist}}).$$

Thus the $\mathbf{x}^{\text{pred}}$-component of the *total* score depends on the (being-denoised) $\mathbf{x}_t^{\text{hist}}$, whereas the prediction-only score uses the marginal $p_t(\mathbf{x}_t^{\text{pred}} \mid \mathbf{c}_t)$.

**Lemma A.3.**  *For any $(\mathbf{x}_t^{pred}, \mathbf{c}_t)$,*

$$\nabla_{\mathbf{x}_t^{pred}} \log p_t(\mathbf{x}_t^{pred} \mid \mathbf{c}_t) = \mathbb{E}\left[\nabla_{\mathbf{x}_t^{pred}} \log p_t(\mathbf{x}_t^{pred} \mid \mathbf{x}_t^{hist}) \,\middle|\, \mathbf{x}_t^{pred}, \mathbf{c}_t\right]. \tag{3}$$

*Proof.* Differentiating $\log p_t(\mathbf{x}_t^{\text{pred}}, \mathbf{c}_t) = \log \int p_t(\mathbf{x}_t^{\text{pred}} \mid \mathbf{x}) \, p_t(\mathbf{x} \mid \mathbf{c}_t) \, d\mathbf{x}$ under the integral and applying Bayes' rule yields equation 3.  □

---

[5] SFdiff trains the total-sequence *conditional* score via DSM and samples with a PC sampler; the DSM equivalence for conditionals is given in Theorem 3.2.

We now compare the reverse SDE drifts (omitting the common diffusion term $g(t)\,d\overline{\mathbf{w}}_t$).

First, **Prediction-only** reverse process in $\mathbf{x}^{\text{pred}}$ takes drift in

$$\dot{\mathbf{x}}_t^{\text{pred}} = \mathbf{f}_{\text{pred}}(t, \mathbf{x}_t^{\text{pred}}) - g(t)^2\, \nabla_{\mathbf{x}_t^{\text{pred}}} \log p_t(\mathbf{x}_t^{\text{pred}} \mid \mathbf{c}_t), \tag{4}$$

with $\mathbf{x}_t^{\text{hist}} \equiv \mathbf{c}_t$ fixed as a (noisy) condition.

On the other hand, **Self-Generation reverse drift** (SFdiff) takes drift in

$$\dot{\mathbf{x}}_t^{\text{hist}} = \mathbf{f}_{\text{hist}}(t, \mathbf{x}_t^{\text{hist}}) - g(t)^2 \Big( \nabla_{\mathbf{x}_t^{\text{hist}}} \log p_t(\mathbf{x}_t^{\text{hist}} \mid \mathbf{c}_t) + \nabla_{\mathbf{x}_t^{\text{hist}}} \log p_t(\mathbf{x}_t^{\text{pred}} \mid \mathbf{x}_t^{\text{hist}}) \Big), \tag{5}$$

$$\dot{\mathbf{x}}_t^{\text{pred}} = \mathbf{f}_{\text{pred}}(t, \mathbf{x}_t^{\text{pred}}) - g(t)^2\, \nabla_{\mathbf{x}_t^{\text{pred}}} \log p_t(\mathbf{x}_t^{\text{pred}} \mid \mathbf{x}_t^{\text{hist}}). \tag{6}$$

Hence, $\mathbf{x}_t^{\text{hist}}$ is *purified on-the-fly* by $\nabla_{\mathbf{x}_t^{\text{hist}}} \log p_t(\mathbf{x}_t^{\text{hist}} \mid \mathbf{c}_t)$, and $\mathbf{x}_t^{\text{pred}}$ uses the increasingly denoised $\mathbf{x}_t^{\text{hist}}$.

**Proposition A.4.** *Under (A2) and the VP drift $\mathbf{f}_{hist}(t, \mathbf{x}) = -\frac{1}{2}\beta(t)\,\mathbf{x}$, consider two runs of equation 5 with the same $\{\mathbf{x}_t^{pred}\}$ and $\mathbf{c}_t$, but different $\mathbf{x}_t^{hist}$ and $\mathbf{x}_t^{hist\,'}$. Then*

$$\frac{d}{dt} \|\mathbf{x}_t^{\text{hist}} - \mathbf{x}_t^{\text{hist}\,'}\|^2 \leq -2m_x g(t)^2 \|\mathbf{x}_t^{\text{hist}} - \mathbf{x}_t^{\text{hist}\,'}\|^2,$$

$$\Rightarrow \quad \|\mathbf{x}_t^{\text{hist}} - \mathbf{x}_t^{\text{hist}\,'}\| \leq \exp\Big(-m_x \int_t^1 g(s)^2\,\mathrm{d}s\Big) \|\mathbf{x}_1^{\text{hist}} - \mathbf{x}_1^{\text{hist}\,'}\|.$$

*Thus $\mathbf{x}_t^{hist}$ contracts exponentially toward the mode/mean of $p_t(\mathbf{x}^{hist} \mid \mathbf{c}_t)$ along the reverse flow.*

*Proof.* Monotonicity of $\nabla_{\mathbf{x}} \log p_t(\mathbf{x} \mid \mathbf{c})$ with parameter $m_x$ controls the symmetric part of the Jacobian of the drift in equation 5; the contribution of $f_{\text{hist}}$ further aids dissipation. The cross-term $\nabla \log p_t(\mathbf{x}_t^{\text{pred}} \mid \mathbf{x}_t^{\text{hist}})$ is 1-Lipschitz in $\mathbf{x}_t^{\text{hist}}$ under (A1) at $\mathbf{x}_t^{\text{pred}}$ and is dominated by $m_x$; the standard Grönwall argument for deriving the solution of ODE gives the stated inequality. $\qquad\square$

We measure robustness by a Lipschitz-type sensitivity of the *output* $\mathbf{x}_0^{\text{pred}}$ with respect to the *condition path* $\mathbf{c}_.$. Fix two noisy conditions $\mathbf{c}_.$ and $\mathbf{c}'_.$, and couple the reverse noise so differences stem only from the drifts.

**Theorem A.5.** *Assume (A1)–(A3) in Appendix A. Let $H(t) := \int_t^1 g(s)^2\,ds$ and $G := H(0) = \int_0^1 g(s)^2\,ds > 0$. Then for noised input condition $\mathbf{c}_s \equiv \mathbf{x}_s^{hist}$, the upper bound of difference between prediction and ground truth $\|\mathbf{x}_0^{pred} - \mathbf{x}_0^{pred\,'}\|$ are*

*(**prediction-only**)* $\quad L \displaystyle\int_0^1 g(s)^2\, \|\mathbf{c}_s - \mathbf{c}'_s\|\, ds \;\leq\; L\,G \sup_{s \in [0,1]} \|\mathbf{c}_s - \mathbf{c}'_s\|,$

*(**total-sequence**)* $\quad L \displaystyle\int_0^1 g(s)^2\, e^{-m_x H(s)} \|\mathbf{c}_s - \mathbf{c}'_s\|\, ds \;=\; \frac{L}{m_x}\big(1 - e^{-m_x G}\big) \sup_s \|\mathbf{c}_s - \mathbf{c}'_s\|.$

*Consequently,*

$$\frac{L}{m_x}\big(1 - e^{-m_x G}\big) \;<\; L\,G \quad \Rightarrow \quad \sup \|\mathbf{x}_0^{pred} - \mathbf{x}_0^{pred\,'}\|_{\text{total-sequence}} \;<\; \sup \|\mathbf{x}_0^{pred} - \mathbf{x}_0^{pred\,'}\|_{\text{pred-only}}.$$

*Thus the* total-sequence *conditional score produces forecasts with strictly smaller sensitivity to condition perturbations than the* prediction-only *score.*

*Proof.* Let $\Delta(t) := \mathbf{x}_t^{\text{pred}} - \mathbf{x}_t^{\text{pred}\,'}$. We couple the two reverse processes (prediction-only or total-sequence) with the *same* reverse Gaussian noise so that pathwise differences arise only from the drift terms

Write the two reverse drifts (prediction-only vs. total) as

$$\dot{\mathbf{x}}_t^{\text{pred}} = \mathbf{f}_{\text{pred}}(t, \mathbf{x}_t^{\text{pred}}) - g(t)^2\, S_t, \qquad \dot{\mathbf{x}}_t^{\text{pred}\,'} = \mathbf{f}_{\text{pred}}(t, \mathbf{x}_t^{\text{pred}\,'}) - g(t)^2\, S'_t,$$

where $S_t$ and $S'_t$ are, respectively,

$$S_t = \begin{cases} \nabla_{\mathbf{x}_t^{\text{pred}}} \log p_t(\mathbf{x}_t^{\text{pred}} \mid \mathbf{c}_t) & \text{(prediction)}, \\ \nabla_{\mathbf{x}_t^{\text{pred}}} \log p_t(\mathbf{x}_t^{\text{pred}} \mid \mathbf{x}_t^{\text{hist}}) & \text{(total)}, \end{cases} \qquad S'_t = \begin{cases} \nabla_{\mathbf{x}_t^{\text{pred}\,'}} \log p_t(\mathbf{x}_t^{\text{pred}\,'} \mid \mathbf{c}'_t) & \text{(prediction)}, \\ \nabla_{\mathbf{x}_t^{\text{pred}\,'}} \log p_t(\mathbf{x}_t^{\text{pred}\,'} \mid \mathbf{x}_t^{\text{hist}\,'}) & \text{(total)}. \end{cases}$$

Subtracting and taking the inner product with the unit vector $\mathbf{u}(t) := \Delta(t)/\|\Delta(t)\|$ (for $\Delta \neq 0$) yields

$$\frac{d}{dt}\|\Delta(t)\| = \mathbf{u}(t)^\top \big(\dot{\mathbf{x}}_t^{\text{pred}} - \dot{\mathbf{x}}_t^{\text{pred}\,'}\big)$$

$$= \mathbf{u}^\top \big(\mathbf{f}_{\text{pred}}(t, \mathbf{x}_t^{\text{pred}}) - \mathbf{f}_{\text{pred}}(t, \mathbf{x}_t^{\text{pred}\,'})\big) - g(t)^2\, \mathbf{u}^\top \big(S_t - S'_t\big).$$

Under VP, $\mathbf{f}_{\text{pred}}(t, \mathbf{y}) = -\frac{1}{2}\beta(t)\mathbf{y}$ contributes a *contractive* term $-\frac{1}{2}\beta(t)\|\Delta(t)\|$. Moreover, by Assumption (A3) ($m_y$-strong convexity in $\mathbf{x}^{\text{pred}}$),

$$\big(\nabla_{\mathbf{x}^{\text{pred}}} \log p_t(\mathbf{x}^{\text{pred}} \mid \cdot) - \nabla_{\mathbf{x}^{\text{pred}}} \log p_t(\mathbf{x}^{\text{pred}\,'} \mid \cdot)\big)^\top (\mathbf{x}^{\text{pred}} - \mathbf{x}^{\text{pred}\,'}) \leq -m_y\|\Delta(t)\|^2,$$

which implies

$$-\mathbf{u}^\top \big(S_t - S'_t\big) \leq -m_y\|\Delta(t)\| + \Gamma_t,$$

with the *forcing* term

$$\Gamma_t := \begin{cases} \big\|\nabla_{\mathbf{x}_t^{\text{pred}}} \log p_t(\mathbf{x}_t^{\text{pred}} \mid \mathbf{c}_t) - \nabla_{\mathbf{x}_t^{\text{pred}}} \log p_t(\mathbf{x}_t^{\text{pred}} \mid \mathbf{c}'_t)\big\|, & \text{(prediction-only)}, \\ \big\|\nabla_{\mathbf{x}_t^{\text{pred}}} \log p_t(\mathbf{x}_t^{\text{pred}} \mid \mathbf{x}_t^{\text{hist}}) - \nabla_{\mathbf{x}_t^{\text{pred}}} \log p_t(\mathbf{x}_t^{\text{pred}} \mid \mathbf{x}_t^{\text{hist}\,'})\big\|, & \text{(total)}. \end{cases}$$

Absorbing the extra $-\frac{1}{2}\beta(t)\|\Delta\|$ (beneficial contraction) into $m_y$ and combining the above,

$$\frac{d}{dt}\|\Delta(t)\| \leq -m_y\, g(t)^2\, \|\Delta(t)\| + g(t)^2\, \Gamma_t. \tag{7}$$

**Case 1: prediction-only.** By the Fisher/mixture identity (Lemma A.3),

$$\nabla_{\mathbf{x}_t^{\text{pred}}} \log p_t(\mathbf{x}_t^{\text{pred}} \mid \mathbf{c}_t) = \mathbb{E}\big[\nabla_{\mathbf{x}_t^{\text{pred}}} \log p_t(\mathbf{x}_t^{\text{pred}} \mid \mathbf{x}_t^{\text{hist}}) \mid \mathbf{x}_t^{\text{pred}}, \mathbf{c}_t\big].$$

Hence, using Jensen and (A1) (Lipschitz in the history argument),

$$\Gamma_t = \Big\|\mathbb{E}\big[\phi_{\mathbf{x}_t^{\text{hist}}}(\mathbf{x}_t^{\text{pred}}) \mid \mathbf{x}_t^{\text{pred}}, \mathbf{c}_t\big] - \mathbb{E}\big[\phi_{\mathbf{x}_t^{\text{hist}}}(\mathbf{x}_t^{\text{pred}}) \mid \mathbf{x}_t^{\text{pred}}, \mathbf{c}'_t\big]\Big\|$$

$$\leq L \cdot \mathsf{W}_1\big(p_t(\mathbf{x}_t^{\text{hist}} \mid \mathbf{c}_t),\, p_t(\mathbf{x}_t^{\text{hist}} \mid \mathbf{c}'_t)\big),$$

where $\mathsf{W}_1$ is the 1-Wasserstein distance. Under the VP channel and strong log-concavity (A2), the posterior map $\mathbf{c}_t \mapsto p_t(\mathbf{x}_t^{\text{hist}} \mid \mathbf{c}_t)$ is 1-Lipschitz in $\mathsf{W}_1$ (e.g., by contraction of Gaussian channels / Brascamp–Lieb), yielding

$$\Gamma_t \leq L\, \|\mathbf{c}_t - \mathbf{c}'_t\|.$$

Plugging into Step 2 and using $e^{-m_y(H(0)-H(s))} \leq 1$,

$$\|\Delta(0)\| \leq \int_0^1 g(s)^2\, L\, \|\mathbf{c}_s - \mathbf{c}'_s\|\, ds \leq L\, G \sup_{s \in [0,1]} \|\mathbf{c}_s - \mathbf{c}'_s\|,$$

which is exactly equation 1.

**Case 2: total-sequence.** Here

$$\Gamma_t = \big\|\phi_{\mathbf{x}_t^{\text{hist}}}(\mathbf{x}_t^{\text{pred}}) - \phi_{\mathbf{x}_t^{\text{hist}\,'}}(\mathbf{x}_t^{\text{pred}})\big\| \leq L\, \|\mathbf{x}_t^{\text{hist}} - \mathbf{x}_t^{\text{hist}\,'}\| \quad \text{by (A1)}.$$

By Proposition A.4 (contractivity of the $\mathbf{x}^{\text{hist}}$-flow under (A2)),

$$\|\mathbf{x}_t^{\text{hist}} - \mathbf{x}_t^{\text{hist}\,'}\| \leq e^{-m_x H(t)}\, \|\mathbf{x}_1^{\text{hist}} - \mathbf{x}_1^{\text{hist}\,'}\|.$$

With the standard synchronous terminal coupling of the reverse SDE, $\mathbf{x}_1^{\text{hist}}$ and $\mathbf{x}_1^{\text{hist}\,'}$ share the same Gaussian noise, so their difference is controlled by the difference of the (forward) conditions; in particular,

$$\|\mathbf{x}_1^{\text{hist}} - \mathbf{x}_1^{\text{hist}\,'}\| \leq \sup_{u \in [0,1]} \|\mathbf{c}_u - \mathbf{c}'_u\|.$$

Therefore

$$\Gamma_t \ \leq \ L\,e^{-m_x\,H(t)}\,\sup_u \|\mathbf{c}_u - \mathbf{c}'_u\|.$$

Plugging this bound into Step 2, we obtain

$$\|\Delta(0)\| \leq L\,\sup_u \|\mathbf{c}_u - \mathbf{c}'_u\| \int_0^1 e^{-m_y(H(0)-H(s))}\,g(s)^2\,e^{-m_x H(s)}\,ds.$$

Since $e^{-m_y(H(0)-H(s))} \leq 1$, it suffices to compute

$$\int_0^1 g(s)^2\,e^{-m_x H(s)}\,ds.$$

Using $H'(s) = -g(s)^2$, the change of variables $u = H(s)$ gives

$$\int_0^1 g(s)^2 e^{-m_x H(s)}ds = \int_{u=G}^0 \left(-e^{-m_x u}\right)du = \int_0^G e^{-m_x u}\,du = \frac{1 - e^{-m_x G}}{m_x}.$$

Hence

$$\|\Delta(0)\| \ \leq \ \frac{L}{m_x}\left(1 - e^{-m_x G}\right)\sup_s \|\mathbf{c}_s - \mathbf{c}'_s\|.$$

Finally, since $1 - e^{-a} < a$ for all $a > 0$, we have

$$\frac{L}{m_x}\left(1 - e^{-m_x G}\right) \ < \ L\,G,$$

which combined with the two bounds proves the strict inequality and completes the proof. $\qquad\square$

## A.2 CONVERGENCE JUSTIFICATION OF SELF-GENERATION

**Theorem A.6.** *For each $L_{SM}^{pred}(\theta)$ and $L_{SM}^{total}(\theta)$, its denoising score matching are represented as follows:*

$$L_{DSM}^{pred}(\theta) = \mathbb{E}_{t,\mathbf{x}^{total},\mathbf{x}_t^{total}}[\lambda(t)||s_\theta(t,\mathbf{x}_t^{pred},\mathbf{x}^{hist}) - \nabla_{\mathbf{x}_t^{pred}}logp(\mathbf{x}_t^{total}|\mathbf{x}^{total})||_2^2]$$

$$L_{DSM}^{total}(\theta) = \mathbb{E}_{t,\mathbf{x}^{total},\mathbf{x}_t^{total}}[\lambda(t)||s_\theta(t,\mathbf{x}_t^{total},\mathbf{x}^{hist}) - \nabla_{\mathbf{x}_t^{total}}logp(\mathbf{x}_t^{total}|\mathbf{x}^{total})||_2^2]$$

*Therefore, these models aim same conditional score function since $\nabla_{\mathbf{x}_t^{total}}logp(\mathbf{x}_t^{total}|\mathbf{x}^{total}) = \nabla_{[\mathbf{x}_t^{hist},\mathbf{x}_t^{pred}]}logp(\mathbf{x}_t^{total}|\mathbf{x}^{total})$.*

**Remark.** Our total-generation loss *recovers*—rather than merely matches—the same conditional score function. Concretely: (i) although SFdiff is trained on the full-sequence score, it still preserves the conditional distribution $p(\mathbf{x}^{pred} \mid \mathbf{x}^{hist})$; (ii) by generating the entire sequence, the model simultaneously denoises the historical part, improving robustness when the input history contains anomalies.

*Proof.* We prove denoising score matching loss of prediction, $L_{DSM}^{total}(\theta)$. The result of $L_{DSM}^{pred}(\theta)$ can be derived similarly. We start from decomposing it:

$$L_{SM}^{total}(\theta) = -2 \cdot \mathbb{E}_t \mathbb{E}_{\mathbf{x}^{hist}} \mathbb{E}_{\mathbf{x}_t^{total}} \langle s_\theta(t,\mathbf{x}_t^{total},\mathbf{x}^{hist}), \nabla_{\mathbf{x}_t^{total}}logp(\mathbf{x}_t^{total}|\mathbf{x}^{hist})\rangle$$
$$+ \mathbb{E}_t \mathbb{E}_{\mathbf{x}^{hist}} \mathbb{E}_{\mathbf{x}_t^{total}} \left[||s_\theta(t,\mathbf{x}_t^{total},\mathbf{x}^{hist})||_2^2\right] + C_1$$

Here, $C_1$ is a constant that does not depend on the parameter $\theta$, and $\langle \cdot, \cdot \rangle$ means the inner product. Then, the first part's expectation of the right-hand side can be expressed as follows:

$$\mathbb{E}_{\mathbf{x}_t^{total}} \langle s_\theta(t,\mathbf{x}_t^{total},\mathbf{x}^{hist}), \nabla_{\mathbf{x}_t^{total}}logp(\mathbf{x}_t^{total}|\mathbf{x}^{hist})\rangle$$

$$= \int_{\mathbf{x}_t^{total}} \langle s_\theta(t,\mathbf{x}_t^{total},\mathbf{x}^{hist}), \nabla_{\mathbf{x}_t^{total}}logp(\mathbf{x}_t^{total}|\mathbf{x}^{hist})\rangle p(\mathbf{x}_t^{total}|\mathbf{x}^{hist})d\mathbf{x}_t^{total}$$

$$= \int_{\mathbf{x}_t^{total}} \langle s_\theta(t,\mathbf{x}_t^{total},\mathbf{x}^{hist}), \frac{1}{p(\mathbf{x}^{hist})} \frac{\partial p(\mathbf{x}_t^{total},\mathbf{x}^{hist})}{\partial \mathbf{x}_t^{total}}\rangle d\mathbf{x}_t^{total}$$

$$= \int_{\mathbf{x}^{total}} \int_{\mathbf{x}_t^{total}} \langle s_\theta(t,\mathbf{x}_t^{total},\mathbf{x}^{hist}), \frac{1}{p(\mathbf{x}^{hist})} \frac{\partial p(\mathbf{x}_t^{total},\mathbf{x}^{hist},\mathbf{x}^{total})}{\partial \mathbf{x}_t^{total}}\rangle d\mathbf{x}_t^{total} d\mathbf{x}^{total}$$

$$= \int_{\mathbf{x}^{total}} \int_{\mathbf{x}_t^{total}} \langle s_\theta(t,\mathbf{x}_t^{total},\mathbf{x}^{hist}), \frac{\partial p(\mathbf{x}_t^{total}|\mathbf{x}^{total}))}{\partial \mathbf{x}_t^{total}}\rangle \frac{p(\mathbf{x}^{hist},\mathbf{x}^{total})}{\mathbf{x}^{hist}} d\mathbf{x}_t^{total} d\mathbf{x}^{total}$$

$$= \int_{\mathbf{x}^{total}} \int_{\mathbf{x}_t^{total}} \langle s_\theta(t,\mathbf{x}_t^{total},\mathbf{x}^{hist}), \frac{\partial p(\mathbf{x}_t^{total}|\mathbf{x}^{total}))}{\partial \mathbf{x}_t^{total}}\rangle p(\mathbf{x}^{total}|\mathbf{x}^{hist})d\mathbf{x}_t^{total} d\mathbf{x}^{total}$$

$$= \mathbb{E}_{\mathbf{x}^{total}} \left[\int_{\mathbf{x}_t^{total}} \langle s_\theta(t,\mathbf{x}_t^{total},\mathbf{x}^{hist}), \frac{\partial p(\mathbf{x}_t^{total}|\mathbf{x}^{total}))}{\partial \mathbf{x}_t^{total}}\rangle d\mathbf{x}_t^{total}\right]$$

$$= \mathbb{E}_{\mathbf{x}^{total}} \left[\int_{\mathbf{x}_t^{total}} \langle s_\theta(t,\mathbf{x}_t^{total},\mathbf{x}^{hist}), \nabla_{\mathbf{x}_t^{total}} \log p(\mathbf{x}_t^{total}|\mathbf{x}^{total})\rangle p(\mathbf{x}_t^{total}|\mathbf{x}^{total})d\mathbf{x}_t^{total}\right]$$

$$= \mathbb{E}_{\mathbf{x}^{total}} \mathbb{E}_{\mathbf{x}_t^{total}} [\langle s_\theta(t,\mathbf{x}_t^{total},\mathbf{x}^{hist}), \nabla_{\mathbf{x}_t^{total}} \log p(\mathbf{x}_t^{total}|\mathbf{x}^{total})\rangle]$$

$$= \mathbb{E}_{\mathbf{x}^{total}} \mathbb{E}_{\mathbf{x}_t^{total}} [\langle s_\theta(t,\mathbf{x}_t^{total},\mathbf{x}^{hist}), \nabla_{\mathbf{x}_t^{total}} \log p(\mathbf{x}_t^{total}|\mathbf{x}^{total})\rangle]$$

The second part's expectation of the right-hand side can be rewritten similarly, therefore we can derive following result:

$$L_{SM}^{total}(\theta) = -2 \cdot \mathbb{E}_t \mathbb{E}_{\mathbf{x}^{total}} \mathbb{E}_{\mathbf{x}_t^{total}} \langle s_\theta(t, \mathbf{x}_t^{total}, \mathbf{x}^{hist}), \nabla_{\mathbf{x}_t^{total}} \log p(\mathbf{x}_t^{total}|\mathbf{x}^{hist}) \rangle$$
$$+ \mathbb{E}_t \mathbb{E}_{\mathbf{x}^{hist}} \mathbb{E}_{\mathbf{x}^{total}} \mathbb{E}_{\mathbf{x}_t^{total}} \left[ \left\| s_\theta(t, \mathbf{x}_t^{total}, \mathbf{x}^{hist}) \right\|_2^2 \right] + C_1$$
$$= L_{DSM}^{total}(\theta) + C_1$$

$C$ is a constant that does not depend on the parameter $\theta$.

Similarly, we compute $L_{DSM}^{pred}(\theta)$. We give proof on inner product part and the other are deduced directly from the case of $L_{DSM}^{total}(\theta)$.

$$\mathbb{E}_{\mathbf{x}_t^{pred}} \langle s_\theta(t, \mathbf{x}_t^{pred}, \mathbf{x}^{hist}), \nabla_{\mathbf{x}_t^{pred}} \log p(\mathbf{x}_t^{pred}|\mathbf{x}^{hist}) \rangle$$

$$= \int_{\mathbf{x}_t^{pred}} \langle s_\theta(t, \mathbf{x}_t^{pred}, \mathbf{x}^{hist}), \nabla_{\mathbf{x}_t^{pred}} \log p(\mathbf{x}_t^{pred}|\mathbf{x}^{hist}) \rangle p(\mathbf{x}_t^{pred}|\mathbf{x}^{hist}) d\mathbf{x}_t^{pred}$$

$$= \int_{\mathbf{x}_t^{pred}} \langle s_\theta(t, \mathbf{x}_t^{pred}, \mathbf{x}^{hist}), \frac{1}{p(\mathbf{x}^{hist})} \frac{\partial p(\mathbf{x}_t^{pred}, \mathbf{x}^{hist})}{\partial \mathbf{x}_t^{pred}} \rangle d\mathbf{x}_t^{pred}$$

$$= \int_{\mathbf{x}^{pred}} \int_{\mathbf{x}_t^{total}} \langle s_\theta(t, \mathbf{x}_t^{pred}, \mathbf{x}^{hist}), \frac{1}{p(\mathbf{x}^{hist})} \frac{\partial p(\mathbf{x}_t^{total}, \mathbf{x}^{hist}, \mathbf{x}^{pred})}{\partial \mathbf{x}_t^{pred}} \rangle d\mathbf{x}_t^{total} d\mathbf{x}^{pred}$$

$$= \int_{\mathbf{x}^{pred}} \int_{\mathbf{x}_t^{total}} \langle s_\theta(t, \mathbf{x}_t^{pred}, \mathbf{x}^{hist}), \frac{\partial p(\mathbf{x}_t^{total}|\mathbf{x}^{total})}{\partial \mathbf{x}_t^{total}} \rangle \frac{p(\mathbf{x}^{hist}, \mathbf{x}^{pred})}{\mathbf{x}^{hist}} d\mathbf{x}_t^{total} d\mathbf{x}^{pred}$$

$$= \int_{\mathbf{x}^{pred}} \int_{\mathbf{x}_t^{total}} \langle s_\theta(t, \mathbf{x}_t^{pred}, \mathbf{x}^{hist}), \frac{\partial p(\mathbf{x}_t^{total}|\mathbf{x}^{total})}{\partial \mathbf{x}_t^{total}} \rangle p(\mathbf{x}^{pred}|\mathbf{x}^{hist}) d\mathbf{x}_t^{total} d\mathbf{x}^{pred}$$

$$= \mathbb{E}_{\mathbf{x}^{pred}} \left[ \int_{\mathbf{x}_t^{total}} \langle s_\theta(t, \mathbf{x}_t^{pred}, \mathbf{x}^{hist}), \frac{\partial p(\mathbf{x}_t^{total}|\mathbf{x}^{total})}{\partial \mathbf{x}_t^{total}} \rangle d\mathbf{x}_t^{total} \right]$$

$$= \mathbb{E}_{\mathbf{x}^{pred}} \left[ \int_{\mathbf{x}_t^{total}} \langle s_\theta(t, \mathbf{x}_t^{pred}, \mathbf{x}^{hist}), \nabla_{\mathbf{x}_t^{total}} \log p(\mathbf{x}_t^{total}|\mathbf{x}^{total}) \rangle p(\mathbf{x}_t^{total}|\mathbf{x}^{total}) d\mathbf{x}_t^{total} \right]$$

$$= \mathbb{E}_{\mathbf{x}^{pred}} \mathbb{E}_{\mathbf{x}_t^{total}} [\langle s_\theta(t, \mathbf{x}_t^{pred}, \mathbf{x}^{hist}), \nabla_{\mathbf{x}_t^{total}} \log p(\mathbf{x}_t^{total}|\mathbf{x}^{total}) \rangle]$$

$\square$

# B  Descriptions of Datasets, Hyperparameters and Miscellaneous Environments

In this section, we describe model architecture, datasets and hyperparameters.

We first describe the diffusion architecture used in SFdiff. To effectively capture the conditional score function along the temporal axis, we adapt DiffWave (Kong et al., 2021) to our settings. Since SFdiff is based on DiffWave, we highlight the key differences. As derived in Theorem 3.2, the input consists of the diffusion timestep, the diffused target data, and historical data, i.e. $t, \mathbf{x}^{\text{hist}}, \mathbf{x}_t^{\text{total}}$. Consistent with previous works (Ho et al., 2020; Kong et al., 2021), the timestep t is embedded into a continuous domain using sinusoidal embedding:

$$embbeding(t) = [\sin(t/N^{0/d}), ..., \sin(t/N^{d-1/d}), \cos(t/N^{0/d}), ..., \cos(t/N^{d-1/d})]$$

, where d is embedding dimension and N is hyperparameterset to 128 and 10,000, respectively. Furthermore, since our main diffusion-based forecasting baselines are DDPM methods, we use VP SDE and an Euler-Maruyama sampling predictor without corrector, which are a generalized formulation of DDPM (c.f. Section 2.1) and a default setting of VP SDE in Song et al. (2020), respectively. All experiments are conducted by using the following software and hardware environments: UBUNTU 18.04 LTS, PYTHON 3.9.12, CUDA 9.1, NVIDIA Driver 470.141, i9 CPU, and GEFORCE RTX 2080 TI.

Table 5: Description of datasets and hyperparameters.

|  | Dimension | Timesteps | Domain | $\gamma$ | $L_{hist}$ | $L_{pred}$ | $N_{step}$ | $N_{iter}$ | $w$ | $N_{iter}^{CFG}$ | # test sample |
|---|---|---|---|---|---|---|---|---|---|---|---|
| Exchange | 8 | 6071 | $\mathbb{R}^+$ | 0.1 | 90 | 30 | 100 | 72 | 0.01 | 36 | 7 |
| Solar | 137 | 7009 | $\mathbb{R}^+$ | 0.1 | 72 | 24 | 200 | 61 | 0.01 | 71 | 7 |
| Electricity | 370 | 5833 | $\mathbb{R}^+$ | 0.5 | 72 | 24 | 50 | 44 | 0.01 | 66 | 7 |
| Taxi | 1214 | 1488 | $\mathbb{N}$ | 0.1 | 48 | 24 | 50 | 14 | 0.1 | 24 | 56 |
| Wiki | 2000 | 792 | $\mathbb{N}$ | 0.1 | 90 | 30 | 250 | 5 | 0.01 | 6 | 5 |

---

**Algorithm 1** Self-Generation training via conditional DSM

---

1: **Inputs:** dataset $\mathcal{D} = \{(\mathbf{x}^{\text{hist}}, \mathbf{x}^{\text{pred}})\}$; VP schedule $\beta(t)$; time sampler $t \sim \text{Unif}(0, 1)$; loss weights $\gamma$.

2: **Model:** score network $s_\theta(t, \mathbf{x}_t^{\text{total}}, \mathbf{x}^{\text{hist}})$ predicting $\nabla_{\mathbf{x}_t^{\text{total}}} \log p_t(\mathbf{x}_t^{\text{total}} \mid \mathbf{x}^{\text{hist}})$, where $\mathbf{x}_t^{\text{total}} = [\mathbf{x}_t^{\text{hist}}, \mathbf{x}_t^{\text{pred}}]$.

3: **Helpers:** VP coefficients $(\alpha_t, \sigma_t)$ s.t. $\mathbf{x}_t = \alpha_t \mathbf{x}_0 + \sigma_t \varepsilon$; masks $M_{\text{hist}}, M_{\text{pred}}$ (same shape as $\mathbf{x}_t^{\text{total}}$) selecting history / future coordinates.

4: **for** each minibatch $(\mathbf{x}^{\text{hist}}, \mathbf{x}^{\text{pred}}) \sim \mathcal{D}$ **do**

5:     Sample $t \sim \text{Unif}(0, 1)$; sample noises $\varepsilon, \eta \sim \mathcal{N}(\mathbf{0}, \mathbf{I})$.

6:     Form clean total sequence $\mathbf{x}^{\text{total}} = [\mathbf{x}^{\text{hist}}, \mathbf{x}^{\text{pred}}]$.

7:     **Forward diffuse the total:** $\mathbf{x}_t^{\text{total}} \leftarrow \alpha_t \mathbf{x}^{\text{total}} + \sigma_t \varepsilon$.

8:     **Score prediction:** $\widehat{\mathbf{s}} \leftarrow s_\theta(t, \mathbf{x}_t^{\text{total}}, \mathbf{x}^{\text{hist}})$.

9:     **DSM target:** For VP, $\mathbf{s}^\star(t, \mathbf{x}_t^{\text{total}}, \mathbf{x}^{\text{hist}}) = -\dfrac{\varepsilon}{\sigma_t}$         {(denoising score target)}

10:     **Weighted loss (history vs. future):**

$$\mathcal{L}(\theta) \leftarrow \gamma \left\| M_{\text{hist}} \odot (\widehat{\mathbf{s}} - \mathbf{s}^\star) \right\|_2^2 + \left\| M_{\text{pred}} \odot (\widehat{\mathbf{s}} - \mathbf{s}^\star) \right\|_2^2.$$

11:     Update $\theta \leftarrow \theta - \eta_\theta \nabla_\theta \mathcal{L}(\theta)$.

12: **end for**

---

## C  CLASSIFIER-FREE GUIDANCE (CFG)

While CFG is a well-known technique in discrete-time DDPMs, we adapt it to the continuous score-SDE framework. We thus provide additional details on how our continuous-time interpretation modifies the standard CFG approach for time-series.

To incorporate an auxiliary classifier in naïve conditional generation, Dhariwal & Nichol (2021) introduced classifier guidance, modifying the standard denoising process by adjusting the estimated noise. Originally, $\epsilon(\mathbf{x}_t|\mathbf{c}) \sim -\sigma_t \nabla_{\mathbf{x}_t} \log p(\mathbf{x}_t|\mathbf{c})$ is replaced with $\tilde{\epsilon}(\mathbf{x}_t|\mathbf{c}) = \epsilon(\mathbf{x}_t|\mathbf{c}) - w\sigma_t \nabla_{\mathbf{x}_t} \log p(\mathbf{c}|\mathbf{x}_t)$, where $w$ is a weighting term, and an additional classifier is trained to calculate $p(\mathbf{c}|\mathbf{x}t)$. From the perspective of score-based SDEs, this approach can be interpreted as altering the score function $\nabla_{\mathbf{x}_t} \log p(\mathbf{x}_t|\mathbf{c})$ to $\nabla_{\mathbf{x}_t} \log \tilde{p}(\mathbf{x}_t|\mathbf{c}) = \nabla_{\mathbf{x}_t} \log p(\mathbf{x}_t|\mathbf{c}) + w\nabla_{\mathbf{x}_t} \log p(\mathbf{c}|\mathbf{x}_t) = \nabla_{\mathbf{x}_t} \log p(\mathbf{x}_t|\mathbf{c}) p(\mathbf{c}|\mathbf{x}_t)^w$, which means $\tilde{p}(\mathbf{x}_t|\mathbf{c}) \sim p(\mathbf{x}_t|\mathbf{c}) p(\mathbf{c}|\mathbf{x}_t)^w$ and effectively incorporating the classifier into the generative process.

To address the dependency on an additional classifier, Ho & Salimans (2022) proposed classifier-free guidance (**CFG**), allowing the generation process to be guided without the need for a separately trained classifier. In CFG, the model learns the modified noise estimate $\tilde{\epsilon}(\mathbf{x}_t|\mathbf{c}) = (1+w)\epsilon(\mathbf{x}_t|\mathbf{c}) - w\epsilon(\mathbf{x}_t)$ by training a single model that handles both conditional and unconditional generations. This is achieved by training with zero-padding for the unconditional case, resulting in $\tilde{\epsilon}_\theta(\mathbf{x}_t, \mathbf{c}) = (1+w)\epsilon_\theta(\mathbf{x}_t, \mathbf{c}) - w\epsilon_\theta(\mathbf{x}_t, \mathbf{0})$.

Generally, applying Classifier-Free Guidance (CFG) to historical data can be seen as a subset of conditional generation, which is a natural concept. However, in many existing conditional diffusion approaches (e.g., text-to-image or time-series forecasting), noisy conditioning inputs can degrade performance, as highlighted by (Na et al., 2024) (see Appendix C of their paper). This is why CFG has rarely been adopted in time-series forecasting methods so far—if the historical data is noisy, naive CFG can amplify that noise and harm predictions, as highlighted in Table 1 of our paper.

In our framework, though, Self-Generation cleans the historical data during the diffusion process, alleviating overdependency on noised historical sequence. Hence, rather than reinforcing noise, CFG strengthens the useful conditional signal. Put differently, we first reduce the noise in, while applying CFG to guide the prediction more effectively. This synergy between Self-Generation and CFG is described in Section 3, where we show that while naive CFG can degrade performance under noisy conditions, our approach remains robust precisely because of the joint denoising mechanism.

## D  EXPLANATIONS ABOUT BASELINES

**Classical Methods.**

- **VAR/VAR-Lasso** (Lütkepohl, 2005): Vector AutoRegression (VAR) estimates linear dependencies across multiple time-series. VAR-Lasso adds an $\ell_1$ penalty to mitigate overfitting in high-dimensional data.

- **GARCH** (van der Weide, 2002): A model that captures time-varying volatility (conditional variance), particularly popular in financial contexts.

- **VES** (Hyndman et al., 2008): A vectorized exponential smoothing method using state-space formulations to handle multivariate trends.

**VAE/State-Space Method.**

- **KVAE** (Fraccaro et al., 2017): Combines a Kalman Filter with a Variational Autoencoder to learn complex latent dynamics for forecasting.

**Deep Learning-Based Methods.**

- **Vec-LSTM (ind-scaling and low-copula)** (Salinas et al., 2019): LSTM-based multivariate forecasting; the "ind-scaling" version assumes independent output distributions, while the "low-copula" version models their joint distribution via copulas.

- **GP scaling/copula** (Salinas et al., 2019): Similar to Vec-LSTM but leverages Gaussian Processes for uncertainty estimation, offering more flexible distributional modeling at higher computational cost.
- **Transformer MAF** (Rasul et al., 2020): A Transformer architecture combined with a Masked Autoregressive Flow for modeling complex, long-range dependencies in multivariate time series.

**Diffusion-Based Methods.**

- **MG-TSD** (Fan et al., 2024): A diffusion-based probabilistic time-series model that uses multi-granularity (fine→coarse) views of the series to guide the denoising process.
- **TimeGrad** (Rasul et al., 2021): A DDPM-based approach that generates forecasts autoregressively, one step at a time.
- **CSDI** (Tashiro et al., 2021): Primarily designed for time-series imputation but can also perform one-shot forecasting of the future horizon by treating it as a masked region.

Unlike these baselines, our **SFdiff** method denoises the entire time-series (both past and future) during generation. By jointly modeling historical and future observations, SFdiff can more effectively handle noisy inputs, leading to improved probabilistic forecasts.

## E WHY GENERATE THE TOTAL SEQUENCE CONDITIONED ON HISTORICAL DATA?

In this section, we theoretically and empirically justify the necessity of conditioning the generation of the total sequence on historical observations. To clearly demonstrate this, we conduct two additional comparisons: 1) total generation without conditioning (unconditional total generation) using replacement methods, and 2) unconditional generation guided by Observation Self-Guidance (OSG) (Kollovieh et al., 2023a) [6]. Both comparisons highlight the benefits and practical advantages of our proposed Self-Generation approach.

**Replacement Method.** Previous approaches (Song et al., 2020; Ho et al., 2022) suggest continuously replacing the historical portion during diffusion with corresponding diffused historical values from the forward SDE. This approach solely utilizes denoised historical information without leveraging original historical data.

Specifically, at denoising step $t$, the history portion of $\mathbf{x}_t^{\text{total}}$ is replaced with forward-diffused historical values. To be specific, Replacement overwrites the history segment ($\mathbf{y}^{\text{hist}}$) with a forward-diffused surrogate ($\mathbf{y}_t^{\text{hist}}$), producing $\tilde{\mathbf{x}}_t = (\mathbf{y}_t^{\text{hist}}, \mathbf{x}_t^{\text{pred}})$ before applying reverse SDE. Equivalently, the actual update obeys

$$\mathrm{d}\tilde{\mathbf{x}}_t = \left[ f - g^2 s_\theta \right](\tilde{\mathbf{x}}_t)\, \mathrm{d}t + g\, \mathrm{d}\bar{\mathbf{w}}_t + \Pi_{\text{hist}}\big( \underbrace{\left[ f - g^2 s_\theta \right]\big(\mathbf{y}_t^{\text{hist}} - \tilde{\mathbf{x}}_t\big)\big)}_{\text{replacement forcing}}\, \mathrm{d}t, \tag{8}$$

where $\Pi_{\text{hist}}$ projects onto the history coordinates. The extra term in equation 8 is an *exogenous, step-dependent* force that (i) violates the Markov consistency of the learned reverse dynamics (the drift now depends on an external process), (ii) makes the reverse trajectory follow a score field conditioned on *mismatched* history (model trained on $(\mathbf{x}_t^{\text{hist}}, \mathbf{x}_t^{\text{pred}})$ but sampled with $(\mathbf{y}_t^{\text{hist}}, \mathbf{x}_t^{\text{pred}})$), and (iii) accumulates bias across steps; the stronger the diffusion noise (earlier $t$), the larger the mismatch and the propagated error.

**Observation Self-Guidance (OSG).** Another method, OSG, guides unconditional generation using score-SDE by computing gradients of the score network (Song et al., 2020). Although OSG improves performance notably (Solar dataset achieved $0.2490 \pm 0.0094$), it requires repeated gradient computations of the score network, resulting in significantly increased computational overhead and limiting practical applicability in high-dimensional settings (Electricity).

---

[6]We omit OSG on our main paper, since SFdiff aims to multivariate generation while OSG targets univariate generation.

OSG add an external force $\nabla_{\mathbf{x}} \log p(\mathbf{y}_{\text{obs}}|\mathbf{x}_t)$ at each reverse step. For a Gaussian observation model with predictor $\hat{y}(x_t)$ and noise covariance $\Sigma$, $\nabla_{\mathbf{x}_t} \log p(\mathbf{y}_{\text{obs}}|\mathbf{x}_t) = J_{\hat{\mathbf{y}}}(\mathbf{x}_t)^{\top}\Sigma^{-1}\big(\mathbf{y}_{\text{obs}} - \hat{\mathbf{y}}(\mathbf{x}_t)\big)$, so the magnitude of the external force scales with the residual $\mathbf{y}_{\text{obs}} - \hat{\mathbf{y}}(\mathbf{x}_t)$. This has two consequences: (i) the per-step, observation-dependent *direct forcing* splits trajectories and can introduce numerical instability (large residuals $\Rightarrow$ stiff drifts, oscillation/divergence); (ii) computing reverse equation requires *additional backpropagations through the score/predictor at every step*, which yields substantial computational and memory overhead in high-dimensional multivariate settings. On a 2-D oscillator with 10% noise we observed MSE exploding from 0.003 to $2.9 \times 10^{11}$ after 5k training steps.

**Why Self-Generation.** Our Self-Generation conditions on the history *once* and denoises the *entire* sequence with the learned score $s_\theta(t, x_t^{\text{total}})$, without step-wise overwriting or external gradient forcing. Hence the reverse process remains Markov-consistent, avoids trajectory splits and bias accumulation from other methods, and matches the runtime of standard sampling (no extra per-step backward passes), yielding both stability and efficiency.

Table 6: Performance comparison of total sequence generation methods: conditioned SFdiff (ours), unconditional generation methods (Replacement and OSG), and prediction-only generation.

| Dataset | SFdiff (ours) | Replacement | OSG | Prediction-only |
|---|---|---|---|---|
| Exchange | **.0054±.0002** | .0057±.0001 | .0055±.0003 | .0063±.0002 |
| Solar | .2501±.0080 | .2774±.0083 | **.2490±.0094** | .2871±.0202 |
| Electricity | **.0153±.0003** | .0173±.0012 | - | .0210±.0013 |

Table 6 summarizes the experimental performance of the unconditional total-generation methods compared to SFdiff (conditioned total generation). Additionally, results of prediction-only generation (presented previously in Table 1 in the main paper) are included for reference.

The results highlight two key observations:

- **Conditioning Improves Forecasting Quality.** Conditioned SFdiff consistently outperforms both unconditional total-generation (Replacement) and prediction-only generation methods across most datasets, while achieving comparable performance to OSG on Solar. This indicates that conditioning on historical data is beneficial, likely because it preserves residual information from the original conditions.

- **Practical Efficiency and Scalability.** Although OSG achieves slightly better performance on the Solar dataset, its requirement for repeated gradient computations makes it computationally expensive and restricts scalability. In contrast, SFdiff directly integrates denoising into the reverse diffusion process, achieving comparable or superior performance across datasets with significantly lower computational overhead.

Overall, these empirical comparisons provide strong justification for conditioning on historical data when generating the total sequence, as our Self-Generation approach effectively balances performance and computational efficiency.

## F    ADDITIONAL QUANTITATIVE EVALUATION ON TOY DATASETS

In this section, we provide additional quantitative evaluation of SFdiff using the toy datasets described in Section 4.1. Specifically, we report the Mean Squared Error (MSE) and Mean Absolute Error (MAE) metrics computed on the prediction segments of the test sets. As shown in Table 7, both MSE and MAE significantly improve with increased $\gamma$, aligning with the visual evidence of purified historical conditions demonstrated in Figure 3. These results further support our claim that total sequence generation is effective in enhancing forecasting robustness under noisy historical conditions.

| Model | Dataset | $\gamma$ | MSE | MAE |
|-------|---------|----------|-----|-----|
| SFdiff | 2D | 0.01 | $.2953 \pm .0000$ | $.3601 \pm .0000$ |
| | | 0.1 | $.0006 \pm .0000$ | $.0147 \pm .0000$ |
| | 3D | 0.01 | $.0210 \pm .0000$ | $.0956 \pm .0000$ |
| | | 0.1 | $.0002 \pm .0000$ | $.0076 \pm .0000$ |

Table 7: Quantitative results on toy datasets. Lower MSE and MAE indicate better forecasting accuracy.

## G    EXTENDED COMPARISON WITH LTSF BASELINES

### G.1    COMPARISON WITHIN OUR ENVIRONMENT

Throughout the main paper we used the standard predictor–corrector (PC) sampler (which we denote "Naive") with optional classifier-free guidance (CFG). Because LTSF methods such as DLinear (Zeng et al., 2022), FEDformer (Zhou et al., 2022) and PatchTST (Nie et al., 2023) are trained via deterministic regression losses, comparing them against our method gives a fairer picture of pure point forecasting quality.

To evaluate, we follow the "moderate-horizon" configuration used 72 historical steps $\rightarrow$ 24-step forecast for SOLAR, ELECTRICITY, TAXI, and WIKI. Metrics are RMSE and MAE (lower is better), which is typical setting of deterministic sampling methods. SFdiff results are averaged over five seeds; $\pm$ indicates one-standard-deviation.

| Model | Dataset | Approach | RMSE | MAE |
|-------|---------|----------|------|-----|
| DLinear | Solar | – | 30.12 | 18.60 |
| | Electricity | – | 409.82 | 49.88 |
| | Taxi | – | 5.07 | 3.34 |
| | Wiki | – | 6887.80 | 1471.84 |
| FEDformer | Solar | – | 30.08 | 19.36 |
| | Electricity | – | 602.59 | 80.57 |
| PatchTST | Solar | – | 31.00 | 18.86 |
| | Electricity | – | 348.36 | 45.28 |
| SFdiff | Solar | CFG | $30.34 \pm 0.12$ | $14.02 \pm 0.04$ |
| | | Naive | $\mathbf{28.53 \pm 0.43}$ | $\mathbf{12.46 \pm 0.13}$ |
| | Electricity | CFG | $\mathbf{321.85 \pm 36.44}$ | $40.39 \pm 0.10$ |
| | | Naive | $326.00 \pm 52.76$ | $\mathbf{40.35 \pm 0.12}$ |
| | Taxi | CFG | $\mathbf{3.95 \pm 0.00}$ | $\mathbf{2.61 \pm 0.00}$ |
| | | Naive | $4.15 \pm 0.00$ | $2.74 \pm 0.00$ |
| | Wiki | CFG | $5910.28 \pm 12.03$ | $\mathbf{698.78 \pm 0.64}$ |
| | | Naive | $\mathbf{5905.92 \pm 70.48}$ | $718.16 \pm 17.88$ |

Table 8: Moderate-horizon forecasting: Naive PC sampler versus applying CFG, plus LTSF regressors. Bold indicates the best among SFdiff variants.

Table 8 underscores that SFdiff remains strong even when evaluated with a fully deterministic sampler, while also highlighting its stability compared to popular regression-style LTSF architectures.

Notably, Robustness in higher dimensions is worth to be mentioned. As dimensionality increases (Electricity $\rightarrow$ Wiki), LTSF models' errors grow quickly, whereas SFdiff's degradation is moderate—confirming our full-sequence denoising advantage. Moreover, Resource constraints became

matter.FEDformer required custom frequency hacks on Electricity and failed on Wiki; PatchTST ran out of GPU memory on several settings. SFdiff, while generative, scales gracefully once trained—it needs no architectural changes for different frequencies or horizons.

## G.2 COMPARISON WITHIN THE LTSF FRAMEWORK

| Dataset | Model | MSE on noise scale $\sigma$ | | | |
|---------|-------|------|------|------|------|
| | | **0.01** | **0.10** | **0.30** | **0.50** |
| Traffic | PatchTST | 1.4121 | 1.6185 | 2.0753 | 2.5314 |
| | DLinear | 0.3999 | 0.6399 | 1.1737 | 1.7077 |
| | **SFdiff** | **0.1253** | **0.1817** | **0.2688** | **0.3386** |
| Electricity | PatchTST | 0.8628 | 1.2109 | 1.9835 | 2.7561 |
| | DLinear | 0.1404 | 0.4171 | 1.0321 | 1.6471 |
| | **SFdiff** | **0.0293** | **0.0616** | **0.1290** | **0.1952** |
| Weather | PatchTST | 0.4023 | 2.9039 | 8.4274 | 13.9479 |
| | DLinear | 0.1240 | 0.2868 | 0.6483 | 1.0096 |
| | **SFdiff** | **0.0317** | **0.0769** | **0.1586** | **0.2337** |

Table 9: Robustness to input corruption on LTSF benchmarks. We add Gaussian noise to the history window at test time and report MSE on the 24-step horizon. SFdiff sustains strong performance across all noise levels.

Based on the above RMSE/MAE comparisons, we further assess robustness on canonical LTSF benchmarks by injecting input noise at test time. We report MSE on prediction length 24 (averaged over five seeds) following common LTSF practice; competitors are PatchTST (state-of-the-art Transformer) and DLinear (strong linear baseline).

For datasets, We selected the three largest Long-Term Time-Series Forecasting (LTSF) benchmarks—Traffic (862 dims), Electricity (321 dims) and Weather (21 climate signals) (please refer to Table 1 of Zeng et al. (2022))—because their dimensionalities exceed or match those in our real-world tests and therefore stress both capacity and robustness. Note that Electricity dataset of LTSF and our normal time-series forecasting are different (in gluonts package, these datasets are distinguished as Electricity and Electricity-nips).

For baselines, we followed common LTSF practice by reporting MSE on the forecast window with prediction length 24 (each number is averaged over five seeds) and compared against PatchTST (state-of-the-art transformer) and DLinear (strong linear baseline). FEDformer was omitted: on these datasets it underperforms DLinear (see Table 6, Appendix D of the paper).

To evaluate noise sensitivity, after training, we added Gaussian corruption to the historical window at four noise levels $\sigma \in \{0.01, 0.1, 0.3, 0.5\}$ ($\sigma$ expressed in data s.d.). When $\sigma$ grown up, the condition became totally different by perturbed noise.

Figure 4 qualitatively corroborates Table 9: even under the most challenging setting ($\sigma = 0.5$), SFdiff maintains stable trajectories, indicating that full-sequence denoising mitigates noise amplification from corrupted conditions.

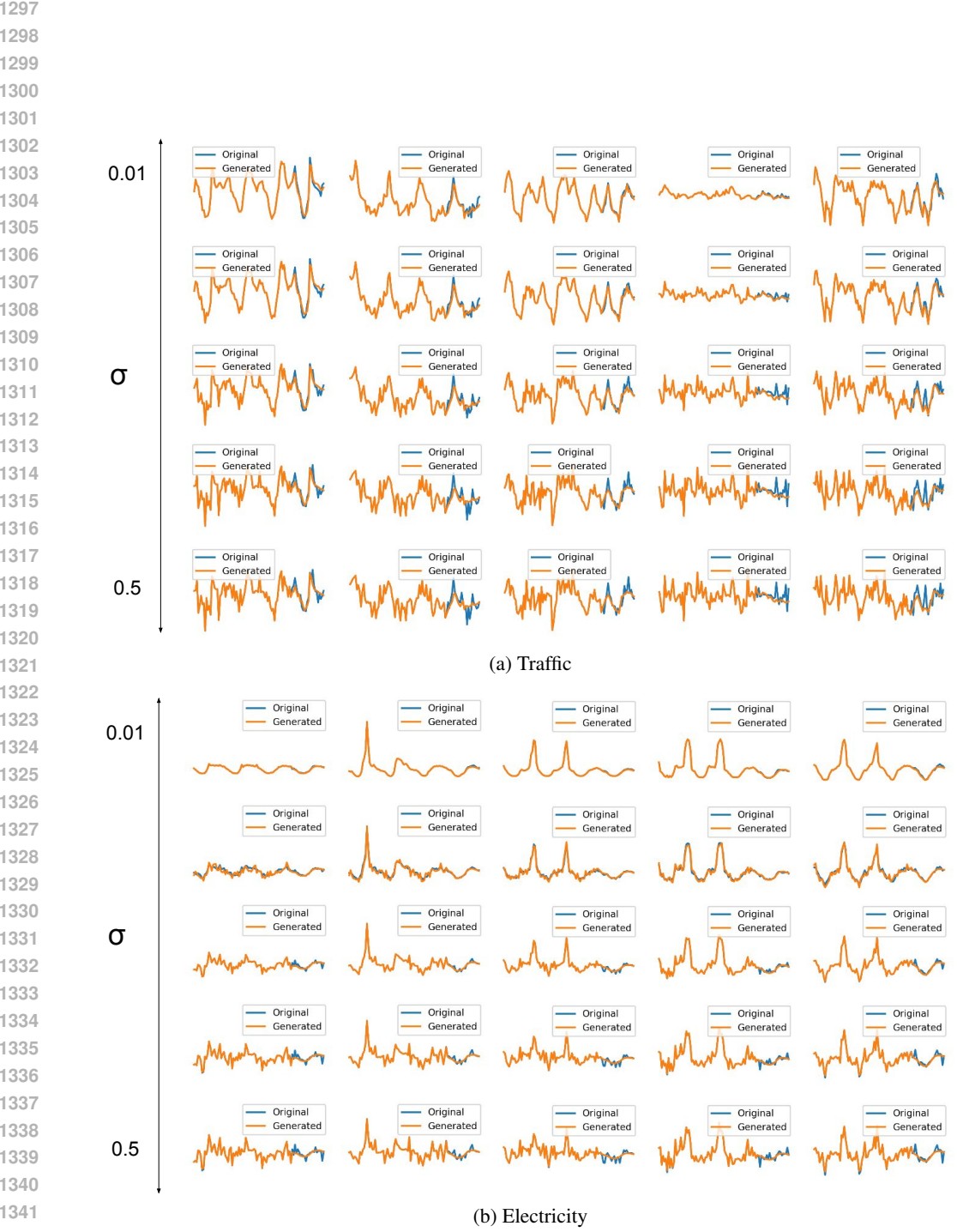

(a) Traffic

(b) Electricity

Figure 4: Representative samples of SFdiff's robustness across increasing injected noise from $\sigma \in \{0.0, 0.01, 0.1, 0.3, 0.5\}$ on TRAFFIC (a) and ELECTRICITY (b).

