# OpenReview forum: "SFdiff : Diffusion Model with Self-Generation for Probabilistic Forecasting"
_ICLR.cc/2026/Conference — Submitted to ICLR 2026_

### Official Review · Reviewer_zyuG · 2025-10-28

**Soundness:** 2
**Presentation:** 2
**Contribution:** 3
**Rating:** 4
**Confidence:** 4

**Summary:**

The paper tackles probabilistic forecasting for multivariate time series using a diffusion-based approach that departs from the common “predict-future-only” setup. The authors propose SFdiff, which reconstructs the entire sequence—past and future—during reverse diffusion. This “self-generation” step implicitly denoises the historical context so that errors and outliers in the conditioning window exert less influence on the generated forecast. The method is formalized with a score-matching objective and a sensitivity bound showing that whole-sequence conditioning is less susceptible to perturbations than future-only conditioning.

**Strengths:**

- **S1** The paper is easy to follow end-to-end, with crisp notation and a clean separation of method, theory, and experiments.
- **S2** The analyses on history/future masking, guidance strength, and sampling steps provide actionable takeaways for reproducing and deploying the method.

**Weaknesses:**

- **W1** The task is time-series forecasting, evaluation relies almost exclusively on CRPS. I would like to see complementary point-forecast metrics (e.g., MAE/MSE/RMSE/SMAPE) to assess accuracy alongside calibration.

- **W2** Qualitative results are shown only for the synthetic setups. It remains unclear how SFdiff behaves visually on real benchmarks. Add forecast trace plots and predictive intervals on several real datasets (e.g., Exchange/Electricity), including challenging noisy cases.

- **W3** Provide sensitivity to key hyperparameters (history/future weight, guidance strength, steps) and statistical significance tests (e.g., paired t-tests or bootstrap CIs) against strong baselines.

- **W4** The framework may benefit from longer contexts (potentially noisier histories), yet the effect of varying input/forecast lengths is not systematically studied. Run controlled studies with longer contexts and horizons to test whether SFdiff’s advantage widens as historical noise increases.

**Questions:**

See W1 to W4.

---

> ### Author Response · Authors · 2025-11-18
>
> We apologize for our mistake in the initial submission, where the Appendix appeared only in the supplementary file.
>
> We now resubmit the full paper **including the Appendix**, with key additions highlighted in blue. Please consider this extended version.
>
> ### **Q1.** The task is time-series forecasting, evaluation relies almost exclusively on CRPS. I would like to see complementary point-forecast metrics (e.g., MAE/MSE/RMSE/SMAPE) to assess accuracy alongside calibration.
>
> Our primary task is **probabilistic forecasting**, so we follow common practice and report **CRPS** as the main metric [1,2,3] because it evaluates the full predictive distribution (calibration \& sharpness). To complement this with point‑forecast accuracy, we now additionally report **MAE/MSE** on the real benchmarks in Appendix G, following LTSF conventions [4,5,6]. The main text cross‑references these tables; CRPS remains the primary score, and point metrics are provided side‑by‑side for completeness.
>
> [1] Rasul et al., Autoregressive Denoising Diffusion Models for Multivariate Probabilistic Time Series Forecasting
>
> [2] Tashiro et al., CSDI: Conditional Score-based Diffusion Models for Probabilistic Time Series Imputation
>
> [3] Rasul et al., Multivariate Probabilistic Time Series Forecasting via Conditioned Normalizing Flows
>
> [4] Zeng et al., Are Transformers Effective for Time Series Forecasting?
>
> [5] Zhou et al., FEDformer: Frequency Enhanced Decomposed Transformer for Long-term Series Forecasting
>
> [6] Nie et al., A Time Series is Worth 64 Words: Long-term Forecasting with Transformers
>
> ### **Q2.** Qualitative results are shown only for the synthetic setups. It remains unclear how SFdiff behaves visually on real benchmarks. Add forecast trace plots and predictive intervals on several real datasets (e.g., Exchange/Electricity), including challenging noisy cases.
>
> We include forecast visualizations under incremental noise on the **LTSF** datasets in Appendix G. For our probabilistic forecasting benchmarks, the per‑series variance is sufficiently high that clean vs. perturbed histories are visually indistinguishable in single‑window plots; therefore, we provide qualitative figures on LTSF (where the contrast is clearer). The LTSF visualizations in Appendix G adequately demonstrate the effect.
>
> ### **Q3.** Provide sensitivity to key hyperparameters (history/future weight, guidance strength, steps).
>
> Sorry for confusion. We describe model architecture and implementation details with hyperparameter setting in Appendix B.
>
> ### **Q4.** The framework may benefit from longer contexts (potentially noisier histories), yet the effect of varying input/forecast lengths is not systematically studied. Run controlled studies with longer contexts and horizons to test whether SFdiff’s advantage widens as historical noise increases.
>
> We are conducting the experiment and will add a **context/horizon sweep** in Appendix G: history length in {96,192}, prediction length in {24,48,96}, $\sigma$ in {0,0.10,0.30,0.50} on Electricity and Traffic in LTSF.
>
> Because the rebuttal window is limited, we are sharing results that are already complete. We are currently running the controlled context/horizon experiments and will include them in Appendix G as soon as they finish.

---

> > ### Comment · Reviewer_zyuG · 2025-11-26
> >
> > Thank you for the efforts made in the revision. However, regarding the newly added experimental results, I notice that DLinear performs substantially better than PatchTST in several settings. This is not entirely consistent with my prior understanding of these models, and it raises the concern that there might be a large discrepancy in hyperparameter choices or other experimental configurations. For the same reason, I also question whether the reported SFDiff results are fully fair and comparable, since the performance gaps among different methods in Table 9 appear unusually large. I would appreciate it if the authors could further clarify this issue and/or provide additional experiments.

---

### Official Review · Reviewer_pypy · 2025-10-28

**Soundness:** 3
**Presentation:** 3
**Contribution:** 3
**Rating:** 4
**Confidence:** 2

**Summary:**

This work presents a robust diffusion framework for probabilistic multivariate time-series forecasting that explicitly addresses noisy or unreliable conditioning histories. Rather than conditioning on a fixed past and generating only future values, the proposed SFdiff jointly reconstructs the past and predicts the future within the same reverse-time process. By doing so, the model “purifies” the historical window on the fly, leading to more stable forecasts. The authors provide a theoretical argument via an upper bound on sensitivity, indicating that the joint sequence score is less affected by perturbations in the history than a future-only score, and supply a compatible denoising score-matching training objective. They further integrate classifier-free guidance into score-based conditional modeling and show that, under self-generation, modest guidance improves calibration and sharpness. Across two synthetic dynamical systems and five real benchmarks, SFdiff generally achieves stronger CRPS_sum than prior diffusion, flow, and non-generative baselines. Comprehensive ablations detail the effect of history/future loss masking, guidance weights, and the number of sampling steps. The approach is promising for scenarios with corrupted contexts, though it requires careful hyperparameter tuning and retains the higher sampling cost typical of diffusion methods.

**Strengths:**

- The toy benchmarks are minimal yet representative, making it straightforward to diagnose where the proposed approach helps under perturbed histories.
- The writing is concise and well structured, allowing readers to grasp the core idea and theoretical claim without unnecessary detours.

**Weaknesses:**

- The ablation study is quite insufficient; for example, I did not see an ablation analyzing the contribution/effect of the historical sequence.
- Since this is a forecasting task, why not include **point-forecast metrics** (e.g., MSE, MAE, RMSE) in the evaluation?
- The model framework is unclear—for example, it is not specified what architecture is used for the denoising network, and many implementation details remain ambiguous.

**Questions:**

See Weakness.

---

> ### Author Response · Authors · 2025-11-18
>
> We apologize for the confusion. In the initial submission the Appendix appeared only in the supplementary file.
>
> We now resubmit the full paper **including the Appendix**, with key additions highlighted in red. Please consider this extended version.
>
> ### **Q1.** Since this is a forecasting task, why not include point-forecast metrics (e.g., MSE, MAE, RMSE) in the evaluation?
>
> Our primary task is **probabilistic forecasting** with relatively short horizons, so we follow standard practice and report **CRPS** as the main metric [1,2,3] because it is a proper scoring rule that evaluates the entire predictive distribution. To address your request for point‑forecast metrics, we additionally report **MSE/MAE** (and RMSE when appropriate) for the LTSF benchmarks in Appendix G, following [4,5,6]. The main text now cross‑references these tables so readers can see both distributional (CRPS) and point‑wise (MSE/MAE) performance.
>
>
> [1] Rasul et al., Autoregressive Denoising Diffusion Models for Multivariate Probabilistic Time Series Forecasting
>
> [2] Tashiro et al., CSDI: Conditional Score-based Diffusion Models for Probabilistic Time Series Imputation
>
> [3] Rasul et al., Multivariate Probabilistic Time Series Forecasting via Conditioned Normalizing Flows
>
> [4] Zeng et al., Are Transformers Effective for Time Series Forecasting?
>
> [5] Zhou et al., FEDformer: Frequency Enhanced Decomposed Transformer for Long-term Series Forecasting
>
> [6] Nie et al., A Time Series is Worth 64 Words: Long-term Forecasting with Transformers
>
> ### **Q2.** The ablation study is quite insufficient; for example, I did not see an ablation analyzing the contribution/effect of the historical sequence.
>
> Thank you for your concern. We agree and have expanded the ablations to isolate the role of the **historical sequence** and its denoising.
>
> We initially described two ablations:
>
> - We set Self‑Generation off (history is fixed to the noisy condition) to obtain a **prediction‑only** variant and compare against full SFdiff, where directly measures the gain from **joint denoising of history+future**.
>
> - We vary (i) the **loss weight** on history $ \gamma$ and (ii) the **CFG strength** $ w $ to show how strongly the model purifies history before forecasting.
>
> Now, we investigate the role of the historical sequence through LTSF baselines and intuitive metrics: MSE and Figures.
>
> - After training, we inject Gaussian noise into the historical window at four levels $ \sigma \in \{0.01, 0.10, 0.30, 0.50\} $ (σ in data s.d.) on **Traffic** (862‑d), **Electricity** (321‑d), and **Weather** (21-d), following LTSF protocol [4]. We compare against PatchTST and DLinear (FEDformer omitted as it underperforms DLinear on these datasets, see Table 9). Full details are in Appendix G.2.
>
> **MSE under noisy history (prediction length = 24, mean over 5 seeds).**
> | Dataset | Model   | σ=0.01 | 0.10   | 0.30   | 0.50   |
> |:--|:--|--:|--:|--:|--:|
> | Traffic | PatchTST | 1.4121 | 1.6185 | 2.0753 | 2.5314 |
> |         | DLinear  | 0.3999 | 0.6399 | 1.1737 | 1.7077 |
> |         | **SFdiff**   | **0.1253** | **0.1817** | **0.2688** | **0.3386** |
> | Electricity | PatchTST | 0.8628 | 1.2109 | 1.9835 | 2.7561 |
> |            | DLinear  | 0.1404 | 0.4171 | 1.0321 | 1.6471 |
> |            | **SFdiff**   | **0.0293** | **0.0616** | **0.1290** | **0.1952** |
> | Weather | PatchTST | 0.4023 | 2.9039 | 8.4274 | 13.9479 |
> |         | DLinear  | 0.1240 | 0.2868 | 0.6483 | 1.0096 |
> |         | **SFdiff**   | **0.0317** | **0.0769** | **0.1586** | **0.2337** |
>
> ### **Q3.** The model framework is unclear—for example, it is not specified what architecture is used for the denoising network, and many implementation details remain ambiguous.
>
> Sorry for confusion. We attach Appendix B describing model architecture and implementation details with hyperparameter setting, reproducibility resources and pseudocode for training.

---

> > ### Comment · Reviewer_pypy · 2025-11-26
> >
> > I would like to thank the authors for their efforts. I acknowledge the progress made in this work, but I also find that there are still several shortcomings. I encourage the authors to conduct more thorough experimental design and reflection. For example, it would be helpful to consider more diverse data noise scenarios. At present, the analysis mainly focuses on the noise inherent in the original sequences, but it remains unclear how to quantitatively characterize the noise level, how SFDiff benefits under different noise regimes, and to what extent it does so. These aspects require more comprehensive experimental validation.
> > In addition, the theoretical analysis of the motivation is solid and logically sound, but I would like to see more direct and intuitive empirical evidence to further strengthen the conclusions.
> > Overall, I believe that my current score is appropriate and will remain unchanged.

---

### Official Review · Reviewer_c6oF · 2025-11-01

**Soundness:** 2
**Presentation:** 2
**Contribution:** 2
**Rating:** 4
**Confidence:** 3

**Summary:**

This paper introduces SFdiff, a diffusion-based model for probabilistic time-series forecasting. The method couples a "Self-Generation" mechanism that jointly denoises past and future segments with classifier-free guidance (CFG), and the experiments span two toy datasets plus five real-world datasets compared against a broad mix of classical and neural baselines.

**Strengths:**

The motivation resonates: noisy historical conditions genuinely hinder diffusion forecasters, and the self-generation story is communicated with helpful illustrations. I also appreciate the wide baseline coverage.

**Weaknesses:**

1. The related-work discussion acknowledges earlier full-sequence diffusion approaches such as TSDiff, yet the manuscript never spells out how SFdiff differs in architecture, loss design, or sampling; despite the γ-sweep and prediction/self-generation comparisons, the incremental novelty over these predecessors remains vague.

2. Theoretical support is also opaque: Theorem 3.1 invokes assumptions (A1–A3) and constants that are never defined, and no proof sketch accompanies the statement, so the promised robustness guarantee cannot be verified.

3. On the empirical side, the narrative that CFG "significantly reduces forecasting errors" conflicts with Table 1/3, where Solar performance worsens once CFG is applied; the text needs to reconcile or explain this divergence.

4. Reproducibility concerns compound the issue: the repeatedly promised "Table 5" with dataset statistics and hyperparameters never appears, leaving key experimental information missing.

**Questions:**

1. What tangible distinctions separate SFdiff from TSDiff and other total-sequence diffusion models, and can ablations quantify the incremental benefit of γ-weighting and self-generation?

2. What exactly are assumptions (A1–A3) and the constants in Theorem 3.1, and how do they connect to the VP-SDE sampler used in practice? A proof or detailed sketch is necessary.

3. Could you supply the missing Table 5 (or an equivalent appendix) that records model architectures, diffusion steps, training schedules, compute budgets, and tuning protocols for every method?

Beyond addressing those questions, it would help to clarify why CFG degrades Solar results while improving others, ideally with diagnostics that measure the claimed purification of historical inputs. Quantitative evidence of the purification effect and high-level pseudocode would also improve the presentation.

---

> ### Author Response · Authors · 2025-11-18
>
> We apologize for the confusion. In the initial submission the Appendix appeared only in the supplementary file.
>
> We now resubmit the full paper **including the Appendix**, with key additions highlighted in blue. Please consider this extended paper.
>
> ### **Q1.** What tangible distinctions separate SFdiff from TSDiff and other total-sequence diffusion models, and can ablations quantify the incremental benefit of $\gamma$-weighting and self-generation?
>
> For detailed explanation, please refer to Appendix E.
>
> **Conceptual distinctions.**
>
>
> * First, *TSDiff/OSG* applies **step‑wise observation‑driven forcing** during unconditional sampling; **SFdiff** learns a **conditional score for the total sequence** and denoises history+future **jointly** without per‑step external gradients.
> * Second, SFdiff’s joint denoising of the history enjoys a **contraction** property (Prop. A.4), which reduces sensitivity to noisy conditions. On the other hand, OSG’s per‑step forcing can become unstable under strong noise and is expensive in high dimensions (extra backward passes each step).
> * Third, SFdiff matches standard sampling (no additional backprop per step). But OSG incurs repeated gradient computations and higher memory footprints; this is prohibitive on multivariate LTSF benchmarks.
>
> **Ablations.**
> *  We share the quantitative results of Figure 3 in Appendix F, sweeping $\gamma\in \{ 0.01,0.1 \}$ and report MSE/MAE. On the noisy 2D/3D datasets, SFdiff preserve robust performance as $\gamma$ increased.
>
> ### **Q2.** What exactly are assumptions (A1–A3) and the constants in Theorem 3.1, and how do they connect to the VP-SDE sampler used in practice? A proof or detailed sketch is necessary.
>
> We give detailed assumptions and connection to VP-SDE in Appendix A. Our assumption built on reasonable background of Diffusion models field: Lipschitz property(A1), concavity(A2) and Coercivity(A3). Also, as you can see from the Proposition A.4, we can enjoy contraction property due to drift term of VP-SDE.
>
> ### **Q3.** Could you supply the missing Table 5 (or an equivalent appendix) that records model architectures, diffusion steps, training schedules, compute budgets, and tuning protocols for every method?
>
> Thank you for your suggestion. We attach model description in Appendix B and Table 5 describing hyperparameter settings for each method and dataset: architecture (layers/hidden sizes/kernels), diffusion steps $T$, sampler settings, compute resources and $\gamma$ selection.
>
> ### **Q4.** On the empirical side, the narrative that CFG "significantly reduces forecasting errors" conflicts with Table 1/3, where Solar performance worsens once CFG is applied; the text needs to reconcile or explain this divergence. Clarify why CFG degrades Solar results while improving others, ideally with diagnostics that measure the claimed purification of historical inputs.
>
> Thank you for your concern. We have revised the text to: **“CFG significantly reduces errors on most datasets, with the only exception of Solar.”** For the reason, we observed Solar is periodic and low‑noise; its predictive signal is already well captured by the base conditional score. CFG on possible noise within history might deteriorate prediction quality. In contrast, Traffic/Electricity are higher‑dimensional and noisier; here CFG improves denoising of the history and yields consistent gains.
>
> ### **Q5.** Quantitative evidence of the purification effect and high-level pseudocode would also improve the presentation
>
> We attach quantative result of Figure 3 with increasing noise scale on toy example in Appendix F. Also, quantitative purification results in Appendix G.2 shows error decrease across injected history noise $\sigma\in\{0.01,0.10,0.30,0.50\}$, showing graceful degradation and superiority to prediction‑only baselines.
>
> Also, we give pseudocode in Appendix B for describing Self‑Generation training (conditional DSM) with the $\gamma$ weighting made explicit.

---

### Official Review · Reviewer_Q7wz · 2025-11-01

**Soundness:** 1
**Presentation:** 2
**Contribution:** 2
**Rating:** 2
**Confidence:** 4

**Summary:**

This paper aims to enhance the performance of conditional diffusion models for time series forecasting, by considering the intrinsic noise within historical context. Specifically, the paper proposed SFDiff that reconstruct the full sequence, including both historical part and the target part, instead of only predicting the target part, given the historical part. The paper claims that, in this way, high-frequency anomalies are largely reduced, and thereby minimizing their impact on forecasting. Experiments over 5 datasets and 12 baselines demonstrate lowest probabilistic forecasting error of the proposed method. Case studies illustrate the anomaly deduction effect. Ablation studies discuss the performance sensitivity against different classifier-free guidance scale and diffusion steps.

**Strengths:**

1. This paper investigates from the perspective that reducing influence of anomalies within look back window by treating them in part of the generation target, which is interesting.
2. The experiments have validated this crucial claim, well supporting the idea that the anomalies can be reduced.
3. The paper is well-structured. Experiments are extensive.

**Weaknesses:**

1. Theorem 3.1 is not rigorous. A tighter upper bound does not necessarily lead to a strictly lower function value. (A1)-(A3) is not mentioned throughout the paper.
2. Baselines are old. The authors should consider newer diffusion-based generation models like TimeDiff, TSDiff, MG-TSD, TMDM, NsDiff, etc.
3. The author should discuss more on the difference and connection between the proposed method and TSDiff, which employs a similar technique named observation self-guidance.
4. Formatting issue: The equations are not numbered. Line 246, 247 out of margin.

**Questions:**

Is the performance gain of the model significantly and positively correlated with how anomalous the dataset is?

---

> ### Author Response · Authors · 2025-11-18
>
> Sorry for the confusion. In the initial submission we mistakenly placed the Appendix only in the supplementary.
>
> We now resubmit the full paper **including the Appendix**, with key additions highlighted in **blue**. Please consider our extended version.
>
> ### **Q1.**  Theorem 3.1 is not rigorous. A tighter upper bound does not necessarily lead to a strictly lower function value.
> (A1)-(A3) is not mentioned throughout the paper
>
> We appreciate the opportunity to clarify.
>
> We do **not** claim that an arbitrary tighter upper bound implies a lower objective. Theorem 3.1 is a **rigorous sensitivity comparison** between prediction‑only and total‑sequence (Self‑Generation) reverse flows under (A1)–(A3). Using Lemma A.2 (conditional factorization) and Lemma A.3 (Fisher/mixture identity), the prediction‑only drift depends on the **posterior over histories** $p_t(\mathbf{x}^{\mathrm{hist}}_t \mid \mathbf{c}_t)$, whereas the total‑sequence drift uses the **current, being‑denoised history** $\mathbf{x}^{\mathrm{hist}}_t$. Proposition A.4 shows the history flow is exponentially contractive under (A2), which leads to different **robustness constants**:
>
> $ \mathbf{(prediction-only)}\quad \||\mathbf{x}^{\mathrm{pred}}_0-\mathbf{x}^{\mathrm{pred}'}_0\|| \le L \int_0^1 g(s)^2 \|\mathbf{c}_s-\mathbf{c}'_s\| \mathrm{d}s \le LG \sup \|\mathbf{c}_s-\mathbf{c}'_s\|$
>
> $ \mathbf{(total-sequence)}\quad \||\mathbf{x}^{\mathrm{pred}}_0-\mathbf{x}^{\mathrm{pred}'}_0\|| \le L \int_0^1 g(s)^2 e^{-m_x H(s)} \|\mathbf{c}_s-\mathbf{c}'_s\|  \mathrm{d}s =  \frac{L}{m_x}\big(1-e^{-m_x G}\big) \sup \|\mathbf{c}_s-\mathbf{c}'_s\|$
>
> where $H(t)=\int_t^1 g(s)^2\,\mathrm{d}s$ and $G=H(0)>0$. Since $1-e^{-a}<a$ for all $a>0$,
>
> $ \frac{L}{m_x}\big(1-e^{-m_x G}\big) < LG.$
>
> Recognizing confusion, we rephrase Section 3 to emphasize:
>
> *Under (A1)–(A3), Self‑Generation has a strictly smaller robustness constant than prediction‑only; thus any Lipschitz evaluation functional inherits a no‑worse (and, under nonzero perturbations, strictly better) worst‑case sensitivity.*
>
> Also, before Theorem 3.1, we add forward pointers to Appendix A, and update the theorem wording to avoid any implication about “arbitrary tighter bounds.”
>
> ### **Q2.** Baselines are old. The authors should consider newer diffusion-based generation models like TimeDiff, TSDiff, MG-TSD, TMDM, NsDiff, etc. Especially, the author should discuss more on the difference and connection between the proposed method and TSDiff, which employs a similar technique named observation self-guidance.
>
> Sorry for confusion. We initially described the case of TSDiff (Observation Self Guidance) in the supplementary material. We placed TSDiff to Appendix since its guidance is designed for **univariate** series and steers **unconditional** sampling by injecting an observation‑driven gradient at **every step**. Based on the original materials, we develop our discussion into the limitation of TSDiff, structure vulnerablity to noisy history. For example, on a 2-D oscillator with 10\% noise, we observed MSE exploding from 0.003 to 2.9×1e11 after 5k training steps. Please refer to Appendix E.
>
> For other baselines, we are sequentially conducting experiments on 2 to 3 recommended models, and We'll report results in the Appendix as soon as possible.
>
> ### **Q3.** Formatting issue: The equations are not numbered. Line 246, 247 out of margin.
>
> Thank you for announcing us. The displayed equations are numbered (or intentionally unnumbered for short displays), and the overfull lines at 246–247 are corrected by switching to `align/split` and tightening line breaks.
>
> ### **Q4.** Is the performance gain of the model significantly and positively correlated with how anomalous the dataset is?
>
> To highlight our performance gain, we include a **toy quantitative study** (App.F) complementing Fig.3, showing that visual purification aligns with improved error under controlled anomaly injection. Although visual purification of Figure 3 doesn't directly means better purification, along with Appendix F, SFdiff's effectiveness becomes more obvious.
>
> Also, in the LTSF stress test (App.G), as history noise $\sigma$ grows, prediction‑only models degrade sharply, whereas **SFdiff** degrades gracefully and remains best in absolute error across Traffic/Electricity/Weather (Table 9).

---

> ### Author Response · Authors · 2025-11-25
>
> Following the suggestion, we have added MG-TSD as a recent diffusion baseline and report its performance in the same $\text{CRPS}_\text{sum}$ setting.
>
> As shown in Table 2, SFdiff and SFdiff-CFG remain competitive or superior: SFdiff achieves the best result on **Solar**, while SFdiff-CFG ties for best on **Electricity** and clearly outperforms all baselines, including MG-TSD, on **Taxi** and **Wiki**.
>
> For TSDiff, we implemented the original observation self-guidance scheme and found it to be highly unstable on noisy multivariate settings (e.g., on a 2-D oscillator with 10% noise, MSE grows from 0.003 to $2.9\times10^{11}$} after 5k steps), which we discuss in detail, together with the conceptual differences to SFdiff’s history-flow guidance, in Appendix E.
>
> **Table 2.** $\text{CRPS}_\text{sum}$ (↓) on evaluation datasets.
>
> | Method               |      Exchange |         Solar |   Electricity |          Taxi |          Wiki |
> | -------------------- | ------------: | ------------: | ------------: | ------------: | ------------: |
> | VES                  |     .005±.000 |     .900±.003 |     .880±.004 |             – |             – |
> | VAR                  |     .005±.000 |     .830±.006 |     .039±.001 |             – |             – |
> | VAR-Lasso            |     .012±.000 |     .510±.006 |     .025±.000 |             – |     3.10±.004 |
> | GARCH                |     .023±.000 |     .880±.002 |     .190±.001 |             – |             – |
> | KVAE                 |     .014±.002 |     .340±.025 |     .051±.019 |             – |     .095±.012 |
> | Vec-LSTM ind-scaling |     .008±.001 |     .391±.017 |     .025±.001 |     .506±.005 |     .133±.002 |
> | Vec-LSTM low-copula  |     .007±.000 |     .319±.011 |     .064±.008 |     .326±.007 |     .241±.033 |
> | GP scaling           |     .009±.000 |     .368±.012 |     .022±.000 |     .183±.395 |     1.48±1.03 |
> | GP copula            |     .007±.000 |     .337±.024 |     .025±.002 |     .208±.183 |     .086±.004 |
> | Transformer MAF      |     .005±.003 |     .301±.014 |     .021±.000 |     .179±.002 |     .063±.003 |
> | **MG-TSD**           |     .007±.002 |     .308±.010 | **.015±.002** |     .116±.013 |     .053±.005 |
> | TimeGrad             |     .006±.001 |     .287±.020 |     .021±.001 |     .114±.020 |     .049±.002 |
> | CSDI                 |     .007±.001 |     .298±.004 |     .017±.000 |     .123±.003 |     .047±.003 |
> | **SFdiff**           |     .006±.000 | **.250±.007** |     .018±.001 |     .122±.001 |     .052±.000 |
> | **SFdiff-CFG**       | **.005±.000** |     .277±.006 | **.015±.000** | **.092±.001** | **.046±.001** |
>
> *(Bold indicates the best score in each column; MG-TSD is bolded on Electricity where it ties with SFdiff-CFG.)*

---

### Meta-Review · Area_Chair_NSai · 2026-01-13

**Summary:**

After reading the reviews and rebuttal, I’m recommending rejection. There’s an appealing intuition here (denoising the history to reduce sensitivity to noisy conditioning), but the work doesn’t rise to a substantial contribution: the novelty relative to prior diffusion-based forecasting/guidance ideas remains murky, and the added experiments feel more like patching gaps than delivering a decisive, well-designed empirical study. Even if some presentation and completeness issues were addressed in the rebuttal, the overall scope and experimental evidence are still too shallow to support the strength of the claims.

**Reviewer Concerns:**

The rebuttal fixes the most concrete blockers (appendix omission, missing hyperparameter table/architecture details, CRPS-only reporting, and the Solar-vs-CFG inconsistency). It also adds at least one newer diffusion baseline (MG-TSD) and a noisy-history stress test that aligns with the paper’s motivation. The concerns that are remain outstanding are: 1/ a crisp novelty story against prior full-sequence diffusion / OSG-style methods, and experimental comparability. The post-rebuttal fairness/tuning skepticism (PatchTST vs DLinear oddities) requires further clearing up, without which, it’s hard to treat the updated tables as decisive evidence rather than “more numbers.”

**Reviewer Scores:**

All reviewers are likely to stay where they are or even lower their scores (e.g. zyuG) due to the new concerns regarding the new results.

---

### Decision · Program_Chairs · 2026-01-26

Reject